# FLI1 promotes IFN-γ-induced kynurenine production to impair anti-tumor immunity

Enni Chen[1,3], Jiawei Wu[1,3], Jiajia Huang[1,3], Wancui Zhu[1], Haohui Sun[1], Xiaonan Wang[1], Dagui Lin[1], Xiaodi Li[1], Dingbo Shi[1], Zhiqiao Liu[1], Jinsheng Huang[1], Miao Chen ®[1] ✉, Fangyun Xie ®[1] ✉ & Wuguo Deng[1,2] ✉

Nasopharyngeal carcinoma (NPC)-mediated immunosuppression within the tumor microenvironment (TME) frequently culminates in the failure of otherwise promising immunotherapies. In this study, we identify tumor-intrinsic FLI1 as a critical mediator in impairing T cell anti-tumor immunity. A mechanistic inquiry reveals that FLI1 orchestrates the expression of CBP and STAT1, facilitating chromatin accessibility and transcriptional activation of IDO1 in response to T cell-released IFN-γ. This regulatory cascade ultimately leads to augmented IDO1 expression, resulting in heightened synthesis of kynurenine (Kyn) in tumor cells. This, in turn, fosters CD8+ T cell exhaustion and regulatory T cell (Treg) differentiation. Intriguingly, we find that pharmacological inhibition of FLI1 effectively obstructs the CBP/STAT1-IDO1-Kyn axis, thereby invigorating both spontaneous and checkpoint therapy-induced immune responses, culminating in enhanced tumor eradication. In conclusion, our findings delineate FLI1-mediated Kyn metabolism as an immune evasion mechanism in NPC, furnishing valuable insights into potential therapeutic interventions.

Nasopharyngeal carcinoma (NPC), a heterogeneous epithelial tumor originating from nasopharyngeal mucosa, manifests predominantly in East and Southeast Asia[1]. Characteristically linked with Epstein-Barr virus (EBV) infection and marked by extensive immune cell infiltration within and surrounding tumor lesions[2], NPC reveals a potential therapeutic landscape where immune checkpoint blockade (ICB) therapies, such as anti-PD-1 agents, may offer significant treatment promise. However, the intricate biology of EBV-associated NPC presents challenges, as these tumors often exploit host immune mechanisms to construct an immunosuppressive TME, thereby thwarting the efficacy of immunotherapies[3–5]. Such immune evasion manifests in diminished response rates, underscoring the pressing need to innovate strategies that mitigate the deleterious impact of NPC on the TME.

CD8+ cytotoxic T cells stand as formidable effectors in the anticancer immune response and constitute the foundation of many contemporary successful cancer immunotherapies. However, a frequently observed and complex phenomenon is the education of cytotoxic T cells by tumor cells, leading to a loss of their functionality[6]. Insight into this process within the context of NPC has been illuminated by previous immunohistochemical (IHC) studies of NPC specimens. These investigations have unveiled an upregulation of certain immune checkpoints, which subsequently induces a state of exhaustion and anergy in cytotoxic T cells, thereby shielding tumor cells from T-cell-mediated destruction[5,7]. This distinctive trait of NPC could hold the key to understanding the mechanisms underlying the failure of immunotherapy in this context. As such, a rigorous and in-depth exploration of these processes could yield essential insights into mitigating the adaptive immune resistance of tumors.

Tumor cells frequently exhibit aberrant activity of transcription factors, enabling them to foster an immunologically "cold" TME and

[1]Sun Yat-sen University Cancer Center, State Key Laboratory of Oncology in South China, Guangdong Key Laboratory of Nasopharyngeal Carcinoma Diagnosis and Therapy, Guangdong Provincial Clinical Research Center for Cancer, Guangzhou, China. [2]Guangdong Provincial Hospital of Integrated Traditional Chinese and Western Medicine, Guangdong, China. [3]These authors contributed equally: Enni Chen, Jiawei Wu, Jiajia Huang. ✉e-mail: chenmiao@sysucc.org.cn; xiefy@sysucc.org.cn; dengwg@sysucc.org.cn

thereby disrupt immune-mediated clearance, gaining a survival and growth advantage[8–10]. This unique aspect presents a distinctive class of drug targets for tumor intervention, with approaches that are regarded as both direct and promising[11]. Indeed, contemporary cancer therapies designed to target transcription factors have been devised and validated as effective through clinical studies[12–14]. As such, interventions aiming at modulating transcription factor activity emerge as potential strategies to enhance cytotoxic T cell function. In the context of our study, we identify FLI1 as a pivotal transcription factor that regulates IFN-γ-induced kynurenine (Kyn) synthesis through a CBP/STAT1-IDO1 axis in tumor cells. Functionally, the tumor-derived Kyn mediates CD8+ T cell exhaustion and regulatory T cell (Treg) differentiation. Importantly, our research reveals that YK-4-279, a pharmacological agent initially developed for Ewing sarcoma[15], can reverse FLI1-mediated suppression of T cell immunity. Furthermore, it synergistically augments the effects of anti-PD-1 therapy in suppressing tumor growth, thereby introducing a potential therapeutic strategy for NPC.

## Results

### Tumor-intrinsic FLI1 promotes CD8+ T cell exhaustion

In a targeted exploration of key tumor transcription factors implicated in cytotoxic T cell exhaustion, we deployed single-sample gene set enrichment analysis (ssGSEA) to classify various tumor types sourced from TCGA and GEO databases. This categorization segmented tumors into high and low levels of T cell exhaustion groups based on the expression of gene sets previously identified as indicators of T cell exhaustion[16,17]. The segregated groups revealed distinct gene expression profiles, leading us to intersect the differentially expressed genes across NPC, colon cancer (COAD), rectal cancer (READ), lung squamous cell carcinoma (LUSC), and skin cutaneous melanoma (SKCM) with a core transcription factor gene set. Subsequently, we identified five tumor transcription factors as indicative of T cell exhaustion (Fig. 1A). Despite bioinformatic analysis revealing the upregulation of these genes, quantitative PCR (qPCR) assessment established that only the transcriptional level of FLI1 was significantly elevated in NPC cell lines and tissues compared to normal controls (Supplementary Fig. S1A, B). This observation was further confirmed by western blot (Supplementary Fig. S1C–F). Moreover, our investigation entailed a comprehensive survival analysis across a spectrum of common malignancies within the TCGA dataset, revealing an inverse association between FLI1 expression and overall survival (OS) (Supplementary Fig. S1G). These findings prompted us to focus our investigation on FLI1.

We next ascertained the role of tumor-intrinsic FLI1 in CD8+ T cell exhaustion by coculturing activated CD8+ T cells with either FLI1-knockout (KO) or wild-type (WT) NPC cells. Strikingly, FLI1 deficiency within tumor cells culminated in decreased PD-1 and TIM-3 expression on CD8+ T cell surfaces, concomitant with increased IFN-γ and TNF-α release by CD8+ T cells (Fig. 1B–E). In vivo analysis, employing the implantation of FLI1 WT and FLI1-KO MC38 cells into C57BL/6 mice, further verified that FLI1-KO tumors manifested conspicuous growth attenuation, as manifested by reduced growth rate and terminal tumor weight (Fig. 1F–H). Importantly, in alignment with in vitro findings, FLI1 KO in tumor cells correlated with a diminished proportion of exhausted CD8+ T cells (Fig. 1I–M). Additional IHC examination of mouse tumors revealed elevated levels of cleaved Caspase-3, with no alteration in Ki-67 levels in FLI1-KO tumors relative to FLI1-WT counterparts (Fig. 1N, O, Supplementary Fig. S1H). Collectively, our findings illuminate tumor-intrinsic FLI1 deficiency as a critical determinant in augmenting CD8+ T cell-mediated tumor clearance.

### FLI1 enhances IFN-γ-IDO1-Kyn axis to mediate T cell immunity suppression

To elucidate the mechanism driving FLI1-mediated CD8+ T cell exhaustion, we conducted RNA sequencing (RNA-seq) on FLI1-KO and WT HK1 cells after co-incubation with activated CD8+ T cells. Differential gene expression analysis revealed a significant enrichment in tryptophan metabolism, as evidenced by KEGG pathway analysis (Fig. 2A). Recognizing that heightened tryptophan metabolism in tumors culminates in increased Kyn synthesis, leading to subsequent T cell dysfunction[7], we examined the FLI1-regulated differentially expressed genes within this pathway. We identified that the expression of Indoleamine 2,3-dioxygenase 1 (IDO1), a pivotal enzyme in Kyn synthesis, was repressed by FLI1 KO (Fig. 2B). Additionally, an analysis of the NPC database (GSE102349) demonstrated a positive correlation between FLI1 and IDO1 mRNA levels (Fig. 2C). Then, we constructed a Kyn metabolism signature using Kyn synthesis-related genes[18]. Bioinformatic analysis revealed a significant association between elevated Kyn metabolism and T cell exhaustion in NPC (Supplementary Fig. S2A). These data collectively indicate that FLI1 potentially augments Kyn production by modulating IDO1 expression to induce CD8+ T cell exhaustion.

IDO1 expression can be influenced upon stimulation of cytokines, such as IFN-γ, interleukin (IL)−6 and prostaglandin E2 (PGE2)[19]. To substantiate the impact of FLI1 on the IDO1-Kyn pathway, we employed western blot and qPCR analyses, revealing that only in the presence of IFN-γ, FLI1 KO resulted in a decline in both mRNA and protein levels of IDO1 in NPC cells (Fig. 2D, E and Supplementary Fig. S2B). Furthermore, an ELISA assay detected a diminished Kyn level in the supernatant of IFN-γ-treated FLI1-KO NPC cells (Fig. 2F). Kyn-mediated T cell dysfunction operates through activating the Aryl hydrocarbon receptor (AHR) in T cells[20], and we correspondingly detected reduced AHR activity in T cells incubated with FLI1-KO cells, as evidenced by mRNA levels of canonical AHR target genes CYP1A1 and CYP1B1[21–23] (Supplementary Fig. S2C). Subsequent rescue experiments demonstrated that stable IDO1 overexpression in FLI1-KO NPC cells mitigated the suppression of Kyn production and AHR activity imposed by FLI1 KO, subsequently aggravating CD8+ T cell exhaustion (Supplementary Fig. S2D−F and Fig. 2G−J). In vivo assays further corroborated these findings, with the blocking of FLI1-IDO1 axis leading to decreased intratumoral Kyn levels (Fig. 2K). Moreover, IDO1 reintroduction negated the suppressive impact of FLI1 KO on tumor growth and T cell exhaustion (Fig. 2L−R).

Kyn is also known to foster Treg differentiation[24], and our bioinformatic investigation further illustrated that both FLI1 and Kyn-associated gene signatures were positively correlated with Treg gene signatures in NPC (Supplementary Fig. S2G, H). Consistent with this, in vitro assays demonstrated that FLI1 deficiency triggered a decrease in CD25+ FOXP3+ cells within CD4+ T cells, which was restored by IDO1 overexpression (Supplementary Fig. S2I). In vivo validation in MC38 mouse xenograft models substantiated the role of the FLI1-IDO1 axis in Treg differentiation (Supplementary Fig. S2J, K). Besides, we applied depleting antibodies against CD8+ and CD4+ T cells to discern the specific contributions of immune modulation to tumor outgrowth. Our results revealed that the depletion of these T cell populations significantly mitigated the tumor-suppressive effects observed with FLI1 deficiency (Supplementary Fig. S2L, S2M). Thus, our comprehensive data illuminate that tumor-intrinsic FLI1 dampens T cell-mediated immunity by potentiating the IFN-γ-IDO1-Kyn pathway, thereby conferring a survival advantage to the tumor.

### FLI1 upregulates CBP to increase promoter accessibility of IDO1

Despite its recognized role as a transcription factor, FLI1 has also been identified as a regulator of chromatin remodeling[25]. To unravel how FLI1 augments IDO1 transcription, we employed chromatin immunoprecipitation followed by sequencing (ChIP-Seq) and assay for transposase-accessible chromatin using sequencing (ATAC-Seq). Following IFN-γ stimulation, we observed a significant reduction in accessible genomic sites in FLI1-deficient cells compared to proficient ones (Supplementary Fig. S3A−C). Of note, the IFN-γ-enhanced

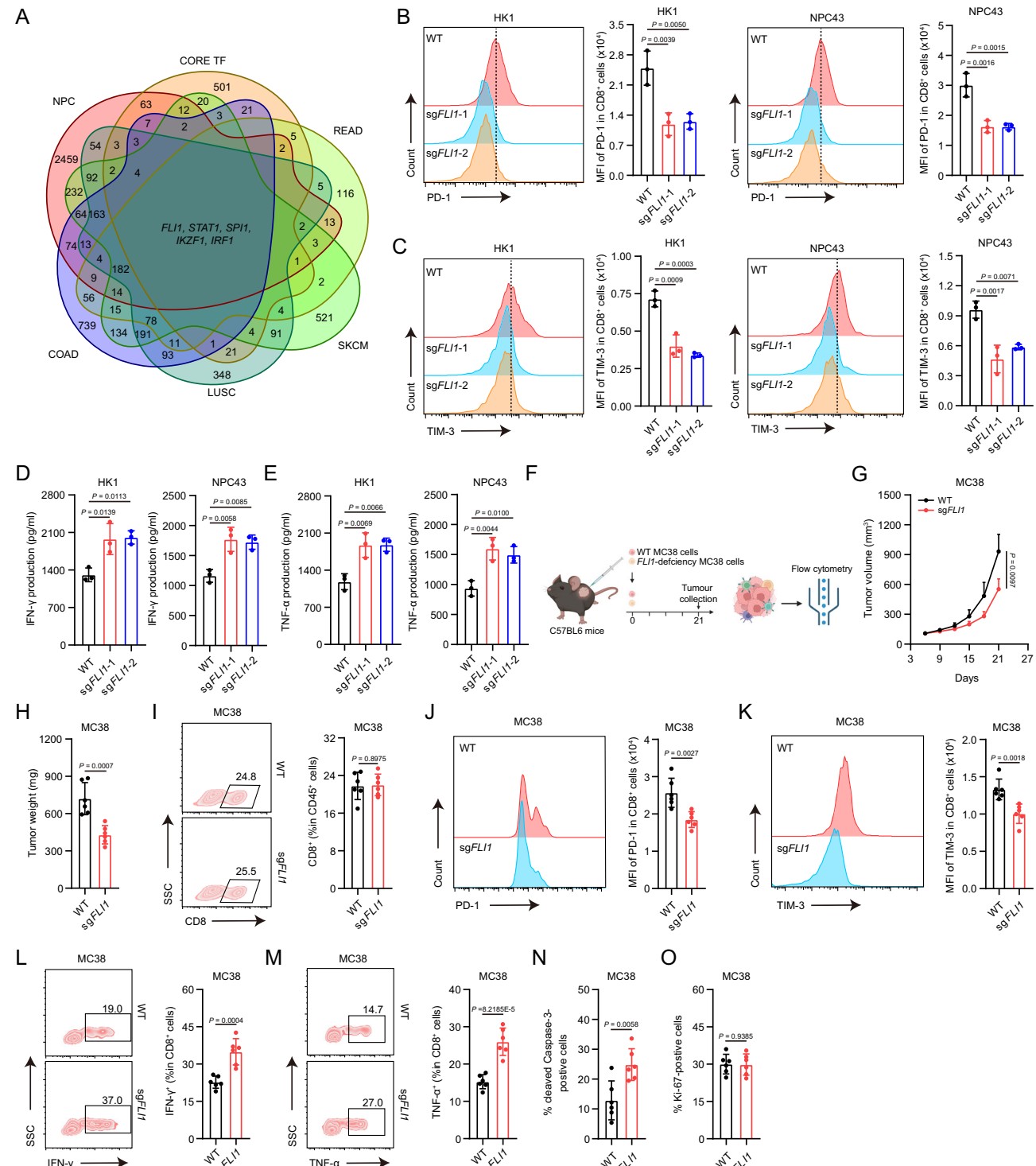

**Fig. 1 | Tumor-intrinsic FLI1 promotes CD8⁺ T cell exhaustion. A** Venn diagram displaying differentially expressed transcription factors between high and low T-cell-exhaustion groups in NPC, COAD, READ, LUSC and SKCM. **B–E** WT and *FLI1*-KO NPC cells were cocultured with activated human CD8⁺ T cells for 48 h. The expression of PD-1 (**B**) and TIM-3 (**C**) on CD8⁺ T cells was determined by flow cytometry. The IFN-γ (**D**) and TNF-α (**E**) levels in culture supernatants were measured by ELISA. Error bars represent the mean ± SD of three independent experiments. Statistical significance was determined using one-way ANOVA with Tukey multiple comparisons test. MFI, mean fluorescence intensity. **F–O** WT or *FLI1*-KO

MC38 cells were inoculated into C57BL/6 mice, and tumors were dissected at day 21 (**F**). The tumor growth rate (**G**) and endpoint tumor weight (**H**) are reported. The percentage of CD8⁺ T cells among CD45⁺ T cells (**I**), the expression of PD-1 (**J**) and TIM-3 (**K**) on CD8⁺ T cells, and the expression of IFN-γ (**L**) and TNF-α (**M**) in CD8⁺ T cells isolated from the indicated tumors were measured by flow cytometry. The percentages of cleaved caspase3- (**N**) and Ki67- (**O**) positive cells from tumors were analyzed by IHC staining. Data (*n* = 6) shown are mean ± SD. Statistical significance was determined using two-way ANOVA (**G**) and two-tailed unpaired *t* test (**H–O**). Source data are provided as a Source Data file.

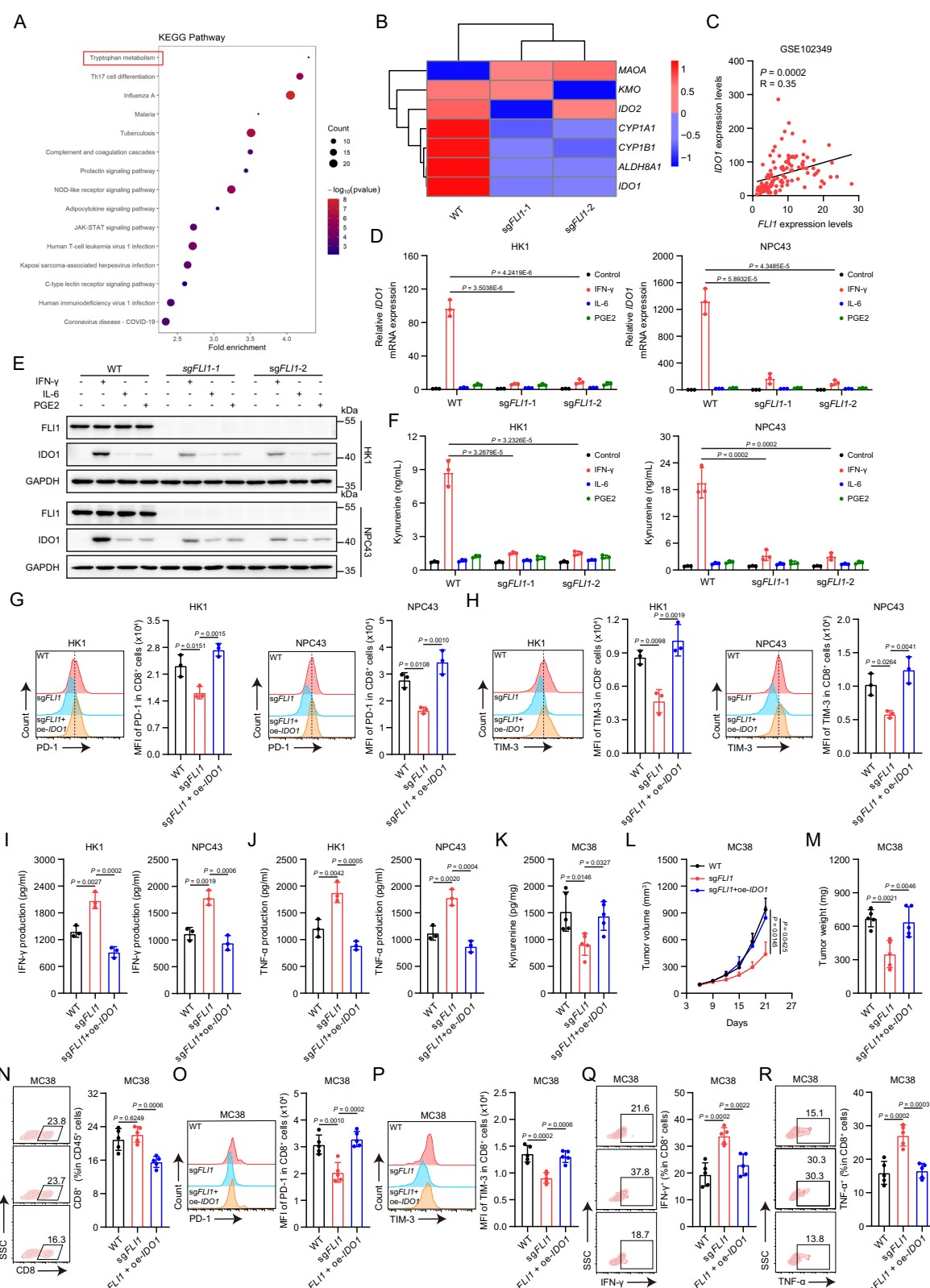

accessibility of *IDO1* promoter was inhibited upon FLI1 loss (Fig. 3A). However, ChIP-seq data revealed no enrichment of FLI1 at the *IDO1* promoter region (Supplementary Fig. S3D), indicating an indirect control of FLI1 over IFN-γ-induced *IDO1* transcription by modulating its promoter accessibility.

　　Building on these insights, we hypothesized that FLI1 might regulate a chromatin modifier, in turn mediating *IDO1* promoter

accessibility. Utilizing the Human TFDB database, we predicted proteins potentially binding to the *IDO1* promoter (Supplementary Fig. S3E), and identified 12 candidates involved in *IDO1* promoter remodeling by overlapping them with a chromatin modifier gene set (Supplementary Fig. S3E). Of these, 10 were found to be transcriptional target genes of FLI1 through our ChIP-seq analysis (Supplementary Fig. S3E). Subsequent qPCR validation showed downregulation of *CBP*

**Fig. 2 | FLI1 enhances IFN-γ-IDO1-Kyn axis to mediate T cell immunity suppression. A** KEGG analysis of differentially expressed genes through RNA-seq following *FLI1* KO after a period of incubation with activated CD8⁺ T cells. When performing enrichment analysis, clusterProfiler primarily uses the hypergeometric test to calculate *P*-values. **B** Heatmap showing differentially expressed genes in tryptophan metabolism pathway. **C** Correlation analysis of the expression of *FLI1* and *IDO1* in NPC. Statistical analysis was conducted by two-sided $\chi^2$ test. **D, E** IDO1 mRNA (**D**) and protein (**E**) levels in *FLI1*-KO and WT HK1 and NPC43 cells upon IFN-γ, IL-6 and PGE2 stimulation were measured by RT-qPCR and western blot. **F** Kyn production in the culture supernatants of *FLI1*-KO and WT HK1 and NPC43 cells upon IFN-γ, IL-6 and PGE2 stimulation was determined by ELISA. **G–J** The indicated NPC cells were cocultured with activated CD8⁺ T cells for 48 h, followed by analyzed with the expression of PD-1 (**G**) and TIM-3 (**H**) on CD8⁺ T cells by flow cytometry.

The IFN-γ (**I**) and TNF-α (**J**) levels in culture supernatants were measured by ELISA. The results are representative of three independent experiments (**D–J**). The data are presented as the mean ± SD. Statistical analysis was performed by one-way ANOVA with Tukey multiple comparisons test (**D, F–J**). **K–R** WT, *FLI1*-KO or *FLI1*-KO + *IDO1*-overexpression (OE) MC38 cells were inoculated into C57BL/6 mice, and tumors were dissected at day 21. The level of Kyn in tumors was assessed by HPLC-MS (**K**). The tumor growth rate (**L**) and endpoint tumor weight (**M**) are reported. The percentage of CD8⁺ T cells among CD45⁺ T cells (**N**), the expression of PD-1 (**O**) and TIM-3 (**P**) on CD8⁺ T cells, and the expression of IFN-γ (**Q**) and TNF-α (**R**) in CD8⁺ T cells isolated from the indicated tumors were measured by flow cytometry. Data (*n* = 5) shown are mean ± SD. Statistical significance was determined using two-way ANOVA (**L**) and one-way ANOVA with Tukey multiple comparisons test (**K, M–R**). Source data are provided as a Source Data file.

and *KAT5* in *FLI1*-KO cells (Supplementary Fig. S3F), corroborated by western blot analysis (Fig. 3B, Supplementary Fig. S3G, H). Notably, only *CBP* knockdown prominently diminished IDO1 mRNA and protein levels in response to IFN-γ (Fig. 3C, D, Supplementary Fig. S3I–L), and concurrently impeded Kyn synthesis (Fig. 3E), and thus establishing CBP as a downstream effector regulating IFN-γ-induced *IDO1* transcription.

To further delineate the regulatory interplay between FLI1, CBP, and IDO1, we performed ChIP-qPCR and luciferase assays. Consistent with ChIP-seq results (Fig. 3F), FLI1 was found to bind to the *CBP* promoter (Fig. 3G), and more importantly, to promote *CBP* transcription (Fig. 3H). CBP-mediated acetylation of histone H3 at lysine residues 9, 14, 18, and 27 is linked to transcriptional activation[26,27]. ChIP-qPCR confirmed that CBP bound to the *IDO1* promoter, mediating acetylation of H3K9 and H3K27, and this effect was amplified by IFN-γ (Fig. 3I, J). Moreover, *FLI1* KO reduced H3K9ac and H3K27ac levels at the *IDO1* promoter under IFN-γ treatment, which were restored by *CPB* overexpression (Fig. 3K). Collectively, our data elucidate a regulatory pathway wherein FLI1 promotes *CBP* transcription, enhancing *IDO1* promoter accessibility, thereby indirectly controlling its transcription.

### FLI1 also regulates STAT1 transcription to mediate IDO1 expression

Overexpression of *CBP* in *FLI1*-KO cells failed to fully restore IDO1 expression and Kyn production (Fig. 3L–N and Supplementary Fig. S3M), pointing to the presence of an alternative mechanism underlying FLI1-mediated *IDO1* transcription. Previous studies have documented STAT1 as a mediator of *IDO1* transcription[28]. Our ChIP-seq data revealed FLI1 occupancy at the *STAT1* promoter (Fig. 4A), and GSEA demonstrated a reduced enrichment of the IFN-γ-induced JAK/STAT signaling pathway in NPC with low *FLI1* expression (Fig. 4B). This led us to hypothesize that FLI1 might also govern *STAT1* transcription to enhance IDO1 expression. We validated this notion using ChIP-qPCR and luciferase assays, confirming the role of FLI1 in transcriptionally regulating STAT1 expression (Fig. 4C, D). Furthermore, qPCR and western blot analyses disclosed a marginal alteration in STAT1 expression under basal conditions, but a marked inhibition following IFN-γ treatment in *FLI1*-KO cells (Fig. 4E, F, Supplementary Fig. S4A, B). This effect was attributed to IFN-γ-induced chromatin remodeling of the *STAT1* gene (Fig. 4G), rendering it more accessible for FLI1-facilitated transcription and thereby modulating STAT1 expression.

To further validate the influence of STAT1 on IDO1 expression in NPC cells, we utilized luciferase assays, which demonstrated that STAT1 indeed mediated *IDO1* transcription (Supplementary Fig. S4C). Additional experiments showed that both IDO1 expression and extracellular Kyn levels were diminished upon *STAT1* knockdown (Supplementary Fig. S4D–G), collectively indicating a transcriptional regulation of the IDO1-Kyn axis by STAT1 in NPC cells. Subsequent rescue experiments through overexpression of *STAT1* alone or both *STAT1* and *CBP* in *FLI1*-KO cells aligned with our hypothesis. While overexpression of *STAT1* alone had negligible effects on IDO1

expression and Kyn production, the concomitant overexpression of *STAT1* and *CBP* completely abrogated the inhibitory consequences of *FLI1* KO on IDO1-mediated Kyn synthesis (Fig. 4H–J and Supplementary Fig. S4H). In summary, our findings reveal a multifaceted mechanism by which FLI1 modulates IDO1 expression at both epigenetic and transcriptional levels through regulation of *CBP* and *STAT* transcription.

### YK-4-279 relieves FLI1-mediated anti-tumor immunosuppression

YK-4-279 is recognized as a functional inhibitor targeting certain ETS family members characterized by the presence of Q336, Y395, D398, and/or K399 amino acid epitopes[15,29]. FLI1 possesses this feature as well, implying that YK-4-279 may act upon FLI1-mediated Kyn production. To validate this potentiality, we constructed a luciferase reporter that expressed a promoter carrying a motif specifically recognized by FLI1 (Fig. 5A). Similar to *FLI1* KO, YK-4-279 significantly decreased the luciferase activity but did not affect FLI1 expression (Fig. 5B and Supplementary Fig. S5A), denoting a direct inhibition of *FLI1* transcriptional activity. We subsequently examined the influence of YK-4-279 on the expression of CBP, STAT1 and IDO1 under IFN-γ treatment, and discerned a reduction in all of these targets in NPC cells (Fig. 5C and Supplementary Fig. S5B–I). As a result, Kyn production was decreased upon YK-4-279 treatment (Fig. 5D). More importantly, T cell AHR activity, CD8⁺ T cell exhaustion and Treg differentiation were remarkably mitigated, when coculturing T cells with NPC cells pretreated by YK-4-279 (Fig. 5E–J). Moreover, apart from its immunomodulatory effects, YK-4-279 also manifested anti-proliferative activity against NPC cells (Supplementary Fig. S5J). Overall, these results support that inhibition of FLI1 transcriptional activity via YK-4-279 relieves Kyn-mediated immune suppression.

### Inhibition of FLI1 transcriptional activity via YK-4-279 enhances ICB sensitivity

Building upon the identified inhibitory effects of YK-4-279 on FLI1-mediated immunosuppressive response, we extended our inquiry into its potential synergistic application with anti-PD-1 therapy for NPC suppression. We established NPC xenografts using *FLI1*-KO or *FLI1*-WT HK1 cells in humanized *NOD/SCID/IL2rγ* null mice (Fig. 6A and Supplementary Fig. S6A). Aligning with findings from MC38 mouse models, tumor intrinsic FLI1 deficiency significantly attenuated tumor growth, intratumoral Kyn production, and the frequency of both exhausted CD8⁺ T cells and Tregs within the TME (Fig. 6B–K). Additionally, administration of YK-4-279 mirrored the effects of *FLI1* KO in TME remodeling to suppress tumor growth (Fig. 6B–K). More importantly, combination of *FLI1* KO or YK-4-279 with anti-PD1 therapy delivered a more pronounced tumor suppression (Fig. 6B, C). This combination therapy further alleviated CD8⁺ T cell exhaustion and Treg differentiation (Fig. 6D–K). These findings were also replicated in MC38 mouse models, further validating the effects of YK-4-279 on improvement of ICB sensitivity (Supplementary Fig. S6B–J). Together,

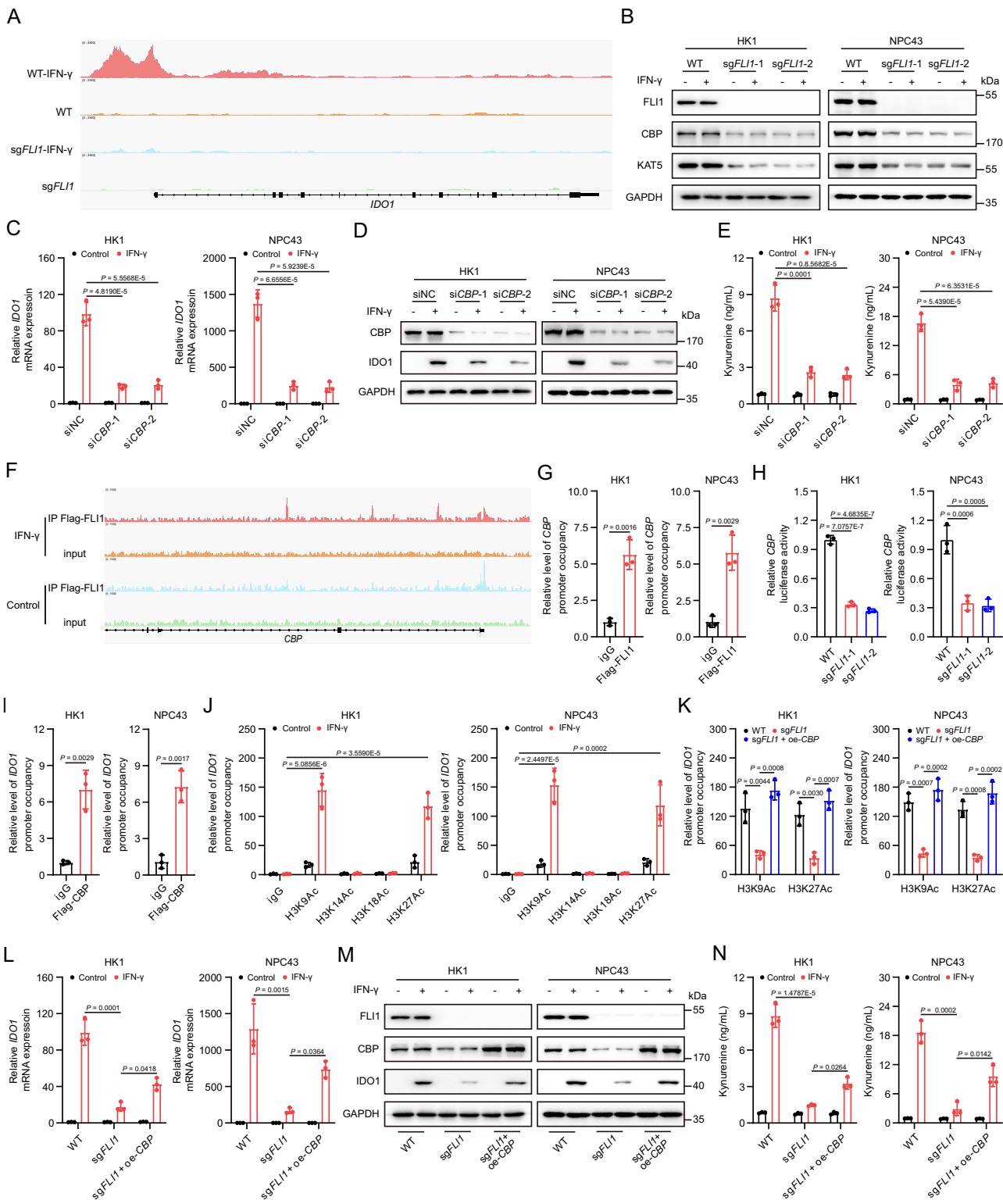

these results illuminate the prospective utility of a combined YK-4-279 and anti-PD-1 therapeutic regimen as an efficacious approach to enhance T cell-mediated anti-tumor immunity, offering a promising avenue for potential advancements in cancer therapy.

## FLI1-IDO1 level serves as an independent predictor of unfavorable prognosis in NPC

To elucidate the clinical significance of FLI1 and IDO1 in NPC, we conducted an IHC analysis on NPC specimens, affirming a positive correlation between FLI1 and IDO1 expression (Fig. 7A, B). Patients were subsequently stratified into high and low FLI1-IDO1 expression groups based on the immunoreactive score (IRS). Our findings revealed that higher FLI1-IDO1 expression was markedly associated with an increased risk of mortality (Fig. 7C). Moreover, Kaplan–Meier analysis and multivariable cox regression revealed that FLI1-IDO1 level had a negative correlation with OS and locoregional recurrence-free survival times (Fig. 7D, E), establishing FLI1-IDO1 expression as an independent prognostic factor in NPC (Fig. 7F, G). Taken together, our

**Fig. 3 | FLI1 upregulates CBP to increase promoter accessibility of IDO1. A** The chromatin accessibility at the *IDO1* promoter in *FLI1*-KO and WT HK1 cells was measured by ATAC-seq. Cells were treated with IFN-γ for 24 h. **B** CBP and KAT5 protein levels in *FLI1*-KO and WT HK1 and NPC43 cells were detected by western blot. **C, D** IDO1 mRNA (**C**) and protein (**D**) levels in *CBP*-knockdown and control HK1 and NPC43 cells were detected. **E** Kyn level in the culture supernatants of *CBP*-knockdown and control HK1 and NPC43 cells was measured. **F** Representation of Flag-FLI1 ChIP-seq and input profiles at the *CBP* gene locus. HK1 cells over-expressing Flag-FLI1 were treated with or without IFN-γ. **G** ChIP-qPCR analysis was performed with an antibody to Flag and *CBP*-promoter-specific primers in HK1 and NPC43 cells overexpressing Flag-FLI1. Cells were treated with IFN-γ for 24 h. **H** *FLI1*-KO and WT HK1 and NPC43 cells were transfected with a *CBP* promoter-luciferase reporter PGL4 plasmid for 24 h. Cells were then treated with IFN-γ for another 24 h, followed by analysis of luciferase activity. **I** ChIP-qPCR analysis was conducted with an antibody to Flag and *IDO1*-promoter-specific primers in HK1 and NPC43 cells overexpressing Flag-CBP, following IFN-γ treatment for 24 h. **J** ChIP-qPCR analysis was carried out with the indicated antibodies (H3K9Ac, H3K14Ac, H3K18Ac and H3K27Ac) and *IDO1*-promoter-specific primers in HK1 and NPC43 cells, with or without IFN-γ treatment. **K** ChIP-qPCR analysis was performed with the indicated antibodies (H3K9Ac and H3K27Ac) and *IDO1*-promoter-specific primers in the indicated NPC cells upon IFN-γ stimulation. **L, M** IDO1 mRNA (**L**) and protein (**M**) levels in the indicated NPC cells with or without IFN-γ treatment were measured. **N** Kyn production in the culture supernatants of the indicated NPC cells was analyzed. The results are representative of three independent experiments (**B–E, G–N**). The data are presented as the mean ± SD (**C, E, G–L, N**). Statistical analysis was performed by one-way ANOVA with Tukey multiple comparisons test (**C, E, H, K, L, N**) and two-tailed unpaired *t* test (**G, I, J**). Source data are provided as a Source Data file.

findings demonstrated the capability of FLI1-IDO1 level for predicting the prognosis in patients with NPC.

## Discussion

NPC is characterized as an "immune-hot" tumor, exhibiting substantial infiltration by tumor-infiltrating lymphocytes (TILs)[30]. Despite this infiltration, tumor cells continue to proliferate, resulting in an immunosuppressive environment within the NPC context. The specific mechanisms driving NPC-mediated TME remodeling remain largely elusive. Herein, we elucidate that Kyn released by NPC cells poses significant barriers to T cell functionality. More importantly, we identify that tumor-intrinsic FLI1 mediates the overproduction of Kyn in response to IFN-γ via CBP/STAT1-IDO1 axis, leading to CD8$^+$ T cell exhaustion and Treg differentiation. Identification of this mechanism would not only refine our understanding of immune-tumor interactions in NPC but could also contribute to the development of more effective immunotherapeutic strategies tailored to overcome these unique challenges.

FLI1, a member of ETS transcription factor family, is highly expressed in both immune cells and multiple cancer cells[31]. It is considered as an oncogene regulating pathways associated with hallmarks of cancers including sustained proliferation, angiogenesis, genomic instability, inhibition of apoptosis and differentiation[32,33]. However, its contribution to the immunomodulatory dynamics within the TME remains enigmatic. A recent study identifies FLI1 of T cell as a factor modulating CD8$^+$ T cell effector differentiation with limited effects on T cell exhaustion[25]. Contrarily, our findings underscore the critical role of tumor-intrinsic FLI1 in promoting T cell exhaustion. This discrepancy potentially attributable to the complex functionality of FLI1 across various tissues. Moreover, the minimal expression of kynurenine pathway genes, such as *IDO1*, *IDO2*, and *TDO2* within T cells, suggests a lack of a FLI1-IDO1 axis in driving their exhaustion. Nonetheless, the identified tumor-intrinsic FLI1-IDO1 axis in our study is profoundly impactful in effectuating TME alteration. Furthermore, evidence shows that CD8$^+$ T cells devoid of FLI1 offer enhanced protection against viral infections and tumors[25]. Given the association between EBV and NPC, FLI1 might modulate a central process in the initiation and progression of NPC, and therefore, present a promising therapeutic target.

Recent investigations pay more attention to IDO1, owing to its significant immunomodulatory impact. IDO1 overexpression as self-protection mechanism of tumor cells has been documented across a spectrum of tumor types[34]. Current researches elucidate this phenomenon predominantly through two regulatory mechanisms: transcriptional induction and post-translational modification. At the transcriptional level, *IDO1* expression is upregulated in response to pro-inflammatory signals, including interferons, tumor necrosis factor-alpha, and various pathogen- and damage-associated molecular pattern. Post-translationally, the aberrant expression of deubiquitinating enzymes such as USP14 in tumors contributes to the stabilization of

IDO1 by impeding its proteasomal degradation[24]. Our data complements that IFN-γ enhances the chromatin accessibility at the *IDO1* locus to facilitate its transcription, and this chromatin remodeling is orchestrated by CBP, which mediates acetylation at H3K9 and H3K27 within the *IDO1* promoter region. Regarding FLI1, its upregulation in NPC promotes *CPB* and *STAT1* transcription, thereby upregulating *IDO1* expression upon IFN-γ stimulation, at both epigenetic and transcriptional levels. These findings not only introduce an effective target for IDO1 interference but also deepen our understanding of the mechanism by which tumor cells architect the immunosuppressive TME through self-reinforcement of metabolic immune checkpoint signaling.

The unique immune landscape of NPC renders immunotherapy suitable for patients with NPC. As such, the incorporation of anti-PD-1 therapy into treatment paradigms has gained clinical prominence, yet it has been found to benefit only 20–30% of NPC patients[30]. Overactive Kyn metabolism, observed in certain tumor cells, has been linked to resistance to immunotherapy and an unfavorable prognosis[35,36]. Previous studies demonstrate elevated expression of IDO1 in NPC samples[5,37], indicating that targeting IDO1 could potentially overcome resistance of NPC to anti-PD-1 therapy. However, despite preclinical evidence supporting the efficacy of IDO1 inhibition in tumor suppression, recent clinical trials have not mirrored these outcomes. This discrepancy may be ascribed to tumors expressing high levels of IL4I1, which produce immunomodulatory metabolites that engage compensatory activation of AHR[38,39]. Another important factor leading to failure in anti-IDO1 therapy is the chemical properties of IDO1 inhibitors. Current small molecule inhibitors of IDO1, predominantly Trp analogs acting as competitive inhibitors, can pose potential off-target effects on compensatory activation of the AHR[19,40]. Our study introduces an innovative strategy for IDO1 inhibition that aims to circumvent the limitations associated with current IDO1 antagonists. YK-4-279 is initially reported as a small molecule targeting the interaction of oncogenic fusion protein EWS-FLI1 with RNA helicases DHX9 and DDX5 in Ewing sarcoma[15]. Interestingly, we observe a potent anti-*FLI1* activity of YK-4-279 in NPC cells. More importantly, YK-4-279 is capable to downregulate IDO1 expression augments T cell immunity and exhibits a synergistic effect with anti-PD-1 therapy. These findings make clinical targeting of FLI1 more attainable and suggesting its potential as a therapeutic intervention for NPC.

In summary, we have deciphered a tumor-intrinsic mechanism of immune tolerance, wherein FLI1 promotes IFN-γ-induced Kyn production, thereby impairing T cell-mediated anti-tumor immunity (Fig. 7H). Based on these findings, the selectively targeting of tumor FLI1 activity emerges as a promising adjuvant strategy in cancer immunotherapy.

## Methods
### Clinical specimens
A total of 110 paraffin-embedded NPC specimens were prospectively acquired to perform a survival analysis. These specimens were

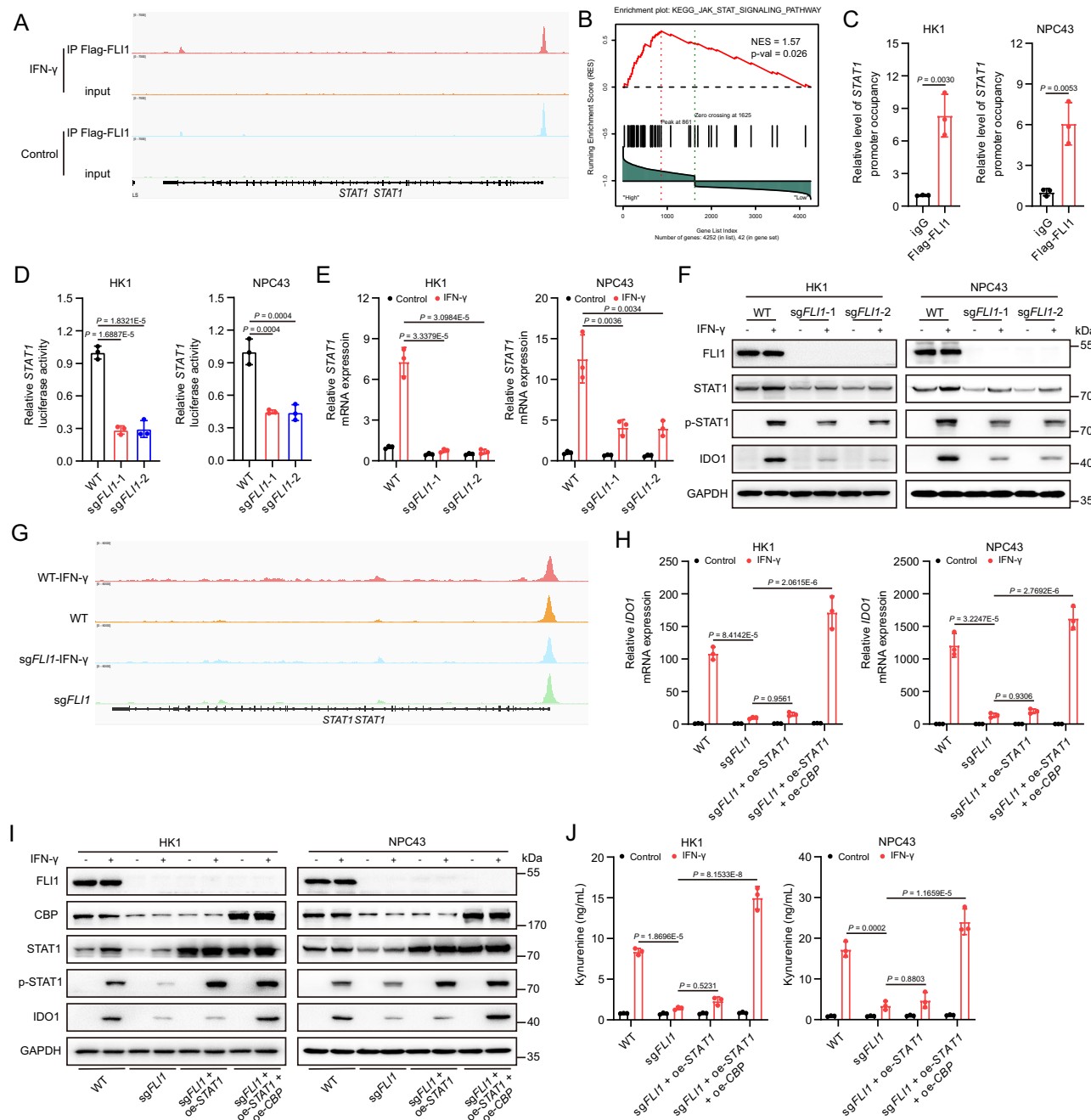

**Fig. 4 | FLI1 also regulates STAT1 transcription to mediate IDO1 expression.** **A** Representation of Flag-FLI1 ChIP-seq and input profiles at the *STAT1* gene locus in HK1 cells overexpressing Flag-FLI1, treated with or without IFN-γ. **B** GSEA from the NPC database (GSE102349) suggested that JAK/STAT signaling pathway was less enriched in NPC with a low *FIL1* expression (two-tailed unpaired *t* test). **C** ChIP-qPCR analysis was conducted with an antibody against Flag and *STAT1*-promoter-specific primers in HK1 and NPC43 cells overexpressing Flag-FLI1, following IFN-γ treatment for 24 h. **D** *FLI1*-KO and WT HK1 and NPC43 cells were transfected with a *STAT1* promoter-luciferase reporter PGL4 plasmid for 24 h. Cells were then treated with IFN-γ for another 24 h, followed by analysis of luciferase activity. **E** *STAT1* mRNA level in *FLI1*-KO and WT HK1 and NPC43 cells, with or without IFN-γ, was determined. **F** STAT1 and p-STAT1 protein levels in FLI1-KO and WT HK1 and NPC43 cells, with or without IFN-γ, were measured. **G** The chromatin accessibility at the *STAT1* gene locus in *FLI1*-KO and WT HK1 cells was measured by ATAC-seq, with or without IFN-γ treatment. **H, I** IDO1 mRNA (**H**) and protein (**I**) levels in the indicated NPC cells, with or without IFN-γ treatment, were determined. **J** Kyn level in the culture supernatants of the indicated NPC cells was detected, with or without IFN-γ treatment. The results are representative of three independent experiments (**C**–**F**, **H**–**J**). The data are presented as the mean ± SD (**C**–**E**, **H**, **J**). Statistical analysis was performed by two-tailed unpaired *t* test (**C**) and one-way ANOVA with Tukey multiple comparisons test (**D**, **E**, **H**, **J**). Source data are provided as a Source Data file.

obtained from Shanghai Outdo Biotech company (Shanghai, China) over the time period from 2010 to 2011. Importantly, all the participants in this investigation had not undergone any form of anti-tumor therapy prior to the collection of the specimens. The tumor

histopathological types were classified following the classification system established by the World Health Organization (WHO). Additionally, the tumor-node-metastasis (TNM) stages were reclassified according to the 8th edition of the American Joint Committee on

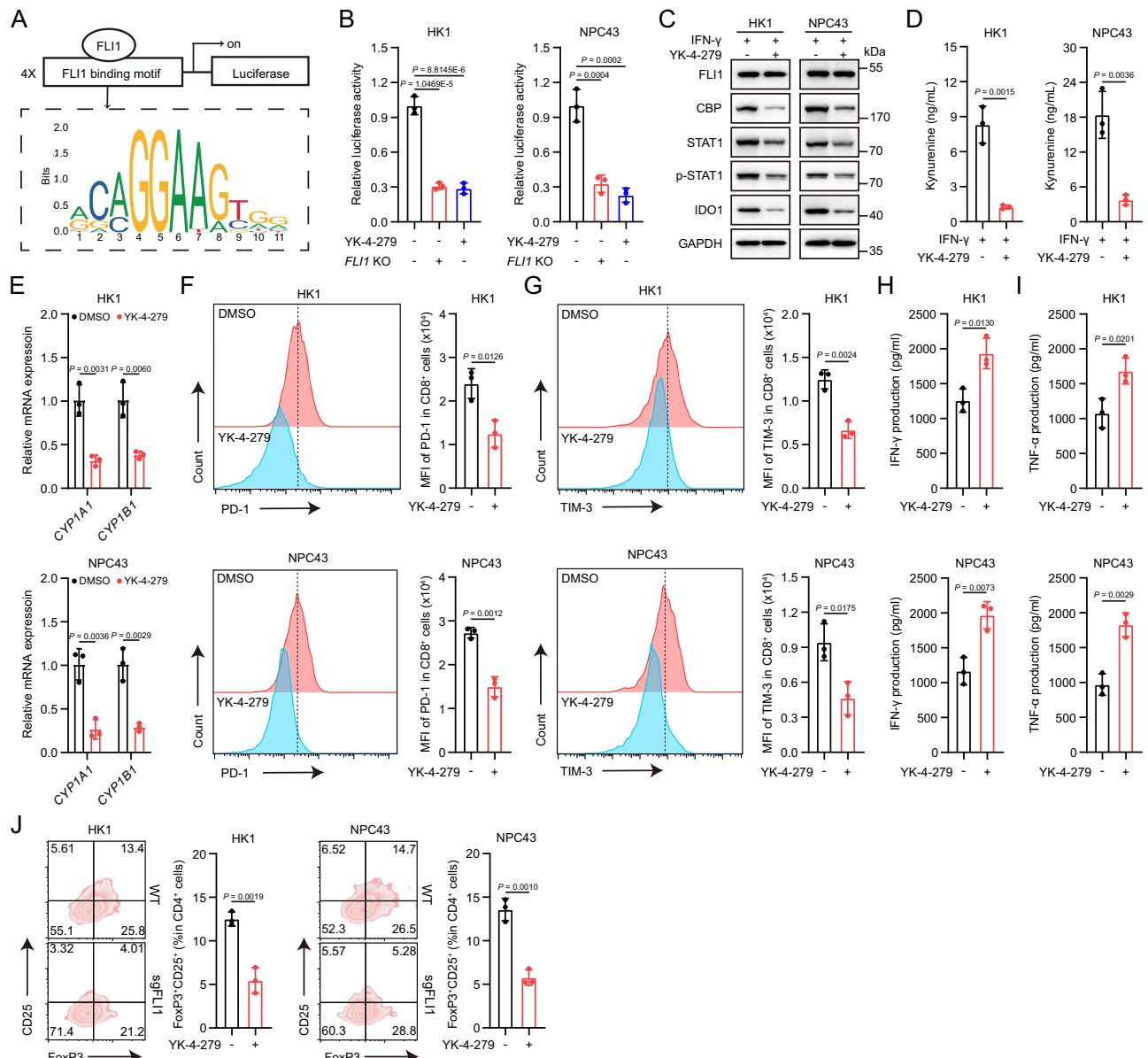

**Fig. 5 | YK-4-279 relieves FLI1-mediated anti-tumor immunosuppression. A** The FLI1-binding motif was obtained from the Jaspar database to construct a luciferase reporter PGL4 plasmid, specifically expressing a promoter carrying the motif recognized by FLI1. **B** HK1 and NPC43 cells were transfected with the promoter-luciferase reporter plasmid for 24 h, then treated with YK-4-279 for an additional 48 h, followed by analysis of luciferase activity. **C, D** HK1 and NPC43 cells were pretreated with either DMSO or YK-4-279 for 24 h, then IFN-γ was added for an additional 24 h. The protein levels of FLI1, CBP, STAT1, p-STAT1 and IDO1 in HK1 and NPC43 cells were measured by western blot (**C**). Kyn production was determined by ELISA (**D**). **E** HK1 and NPC43 cells were pretreated with either DMSO or YK-4-279 for 24 h, then cocultured with activated human T cells for an additional 48 h. *CYP1A1* and *CYP1B1* mRNA levels in T cells were detected. **F–I** HK1 and NPC43 cells were

pretreated with either DMSO or YK-4-279 for 24 h, then cocultured with activated human CD8+ T cells for an additional 48 h. The expression of PD-1 (**F**) and TIM-3 (**G**) on CD8+ T cells was determined by flow cytometry. The IFN-γ (**H**) and TNF-α (**I**) levels in the culture supernatants were measured by ELISA. **J** HK1 and NPC43 cells were pretreated with either DMSO or YK-4-279 for 24 h, then cocultured with activated human CD4+ T cells for an additional 48 h. The proportion of CD25+ FoxP3+ cells among CD4+ T cells was determined by flow cytometry. The results are representative of three independent experiments (**B–J**). The data are presented as the mean ± SD (**B, D–J**). Statistical analysis was performed by one-way ANOVA with Tukey multiple comparisons test (**B**) and two-tailed unpaired *t* test (**D–J**). Source data are provided as a Source Data file.

Cancer (AJCC) Cancer Staging Manual. As a component of their treatment protocol, all patients received radical radiotherapy and platinum-based chemotherapy. Comprehensive clinical characteristics of the patients are presented in Supplementary Table 1. Informed consent was obtained from all patients and approved by the research medical ethics committee of Shanghai Outdo Biotech company (Approval Number: SHYJS-CP-1704009). The NPC specimens used for RNA extraction were obtained from the Sun Yat-sen University Cancer Center (Guangzhou, China). The Institutional Ethical Review Boards of

Sun Yat-sen University Cancer Center approved this study (B2023-232-01).

### Animal experiments

Animal experiments in this study were approved by the Experimental Animal Ethics Committee, Sun Yat-sen University Cancer Center (L102042023050F). 6-week-old female C57BL/6 mice were procured from the Guangdong Medical Laboratory Animal Center (Foshan, China). The mice were subsequently housed in a controlled environment

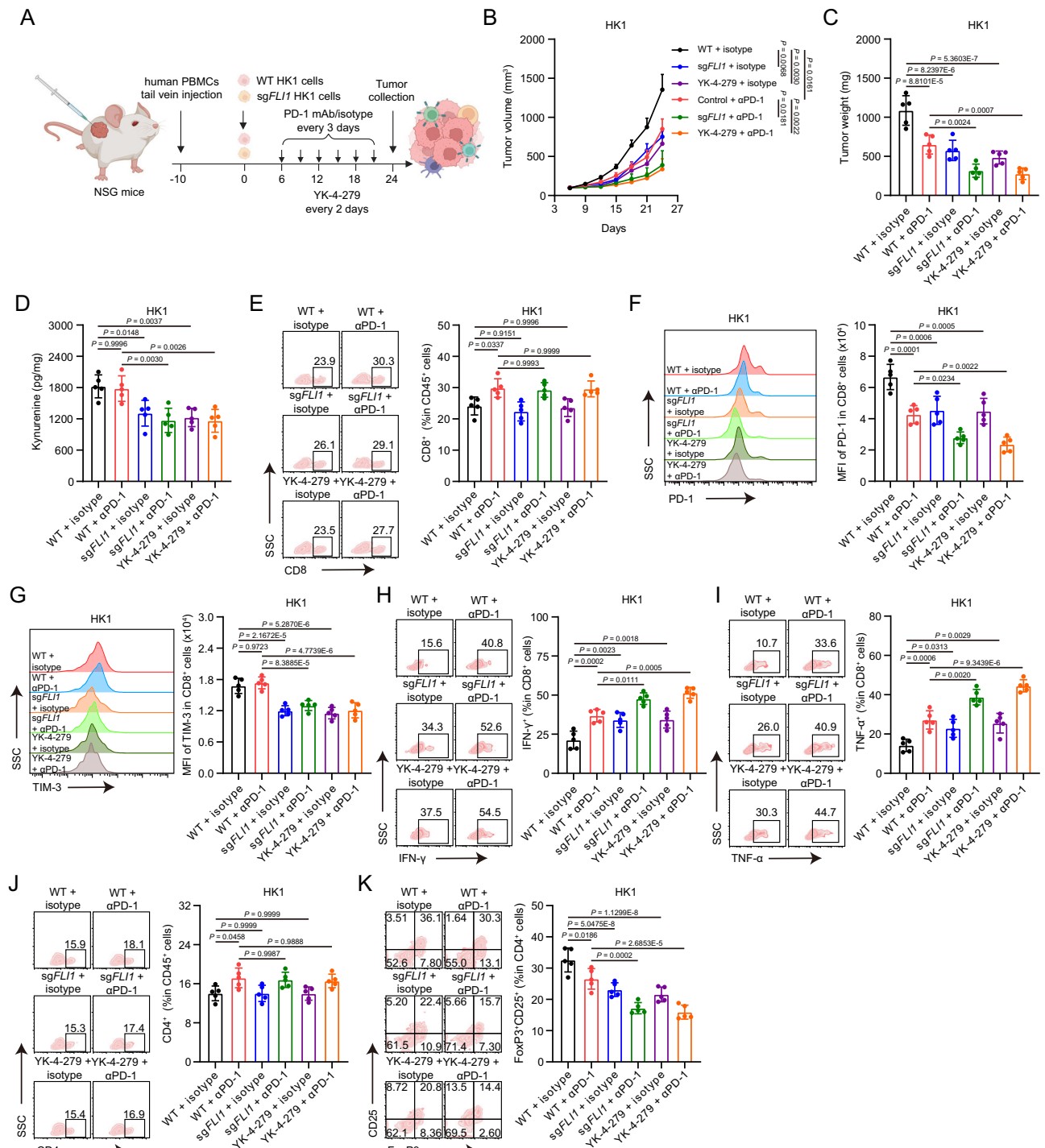

**Fig. 6 | Inhibition of FLI1 transcriptional activity via YK-4-279 enhances ICB sensitivity. A** Schematic of tumor inoculation and treatment in humanized NSG mice. **B–K** WT or FLI1-KO HK1 cells were inoculated into C57BL/6 mice, and tumors were dissected at day 24. The tumor growth rate (**B**) and endpoint tumor weight (**C**) are reported. The level of Kyn in tumors was assessed by HPLC-MS (**D**). The percentage of CD8+ T cells among CD45+ T cells (**E**), the expression of PD-1 (**F**) and TIM-3 (**G**) on CD8+ T cells, and the expression of IFN-γ (**H**) and TNF-α (**I**) in CD8+ T cells isolated from the indicated tumors were measured by flow cytometry. The percentage of CD4+ T cells among CD45+ T cells (**J**) and the proportion of CD25+ FoxP3+ cells among CD4+ T cells (**K**) isolated from the indicated tumors were measured by flow cytometry. Data (*n* = 5) shown are mean ± SD. Statistical significance was determined using two-way ANOVA (**B**) and one-way ANOVA with Tukey multiple comparisons test (**C–K**). Source data are provided as a Source Data file.

at the Animal Experiment Center of Sun Yat-Sen University, ensuring specific pathogen-free conditions. C57BL/6 mice were subcutaneously injected with WT, *FLI1*-KO or *FLI1*-KO + *IDO1*-OE MC38 cells at a density of 1 × 10^6 cells and sacrificed on day 21. The anti-CD8α antibody (100 μg per mouse, BioXcell, BE0061) and anti-CD4 antibody (100 μg per mouse, BioXcell, BE0119) were administered intraperitoneally every 3 days. In addition, the mice were also subcutaneously injected with MC38 cells at a density of 1 × 10^6 cells. On the day 6, the mice were randomly assigned to treatment with anti-PD-1 (200 ug/dose, BioXcell, BE0146), YK-4-279 (60 mg/kg, Selleck, S7679), anti-PD-1 plus YK-4-279, or isotype control, and the mice were sacrificed on day 24. The weights of the excised tumors were documented and recorded.

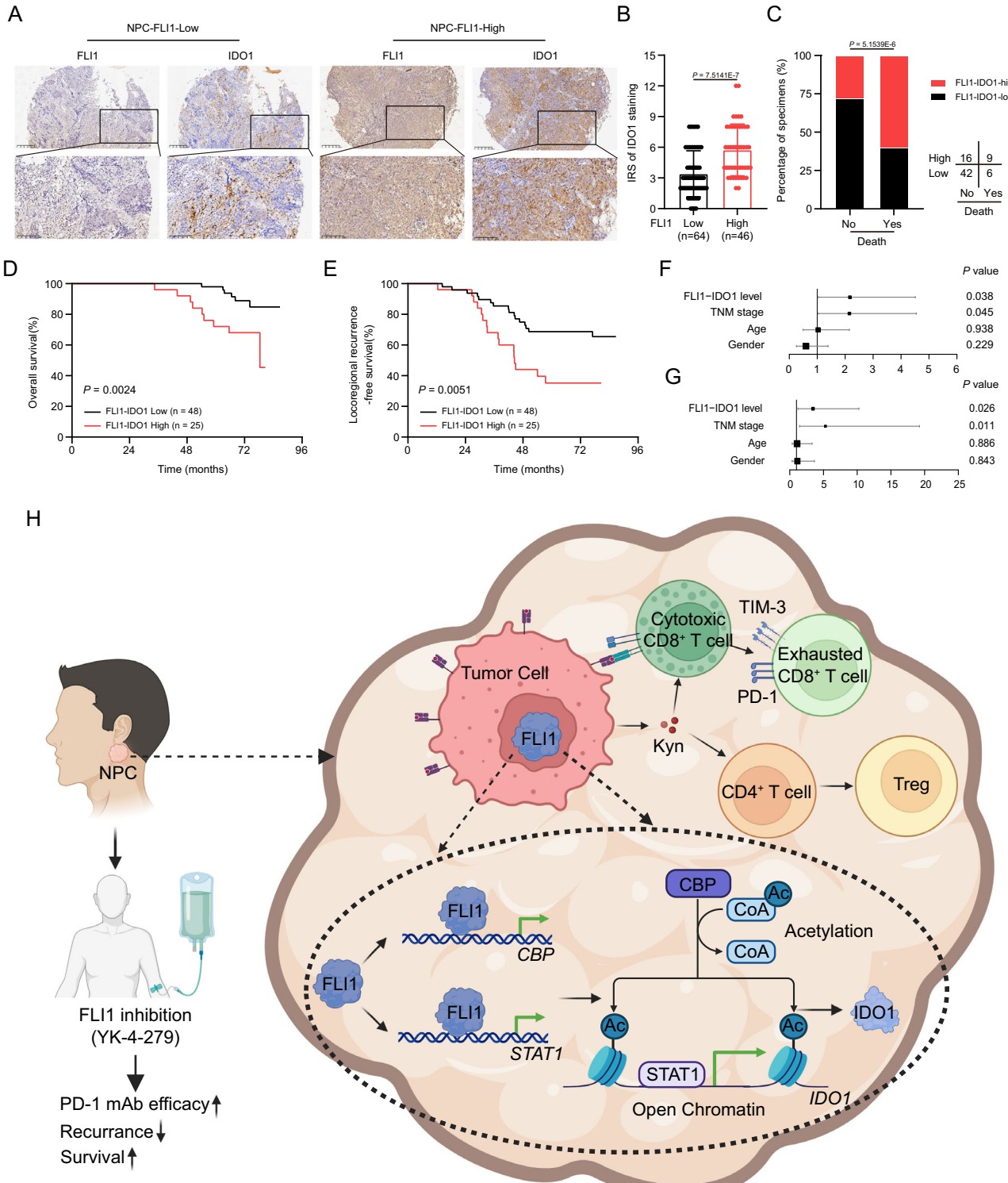

**Fig. 7 | FLI1-IDO1 level serves as an independent predictor of unfavorable prognosis in NPC. A** FLI1 and IDO1 protein expression was evaluated by IHC staining in 110 NPC tumor tissues. Scale bars, 200 μm (upper panel), 100 μm (lower panel). **B** IDO1 IHC scores in NPC tissues with high and low FLI1 expression (*n* = 110). The data are presented as the mean ± SD. Statistical analysis was performed by two-tailed Mann−Whitney test. **C** The association between FLI1 expression and survival status in NPC samples. Statistical analysis was conducted by two-sided $\chi^2$ test. **D**−**G** Kaplan−Meier analysis of overall (**D**) and locoregional recurrence-free (**E**) survival based on the FLI1-IDO1 expression level (*n* = 73). Statistical analysis was performed by log-rank test. Forest plots showing the significance of different prognostic variables in NPC overall (**F**) and locoregional recurrence-free (**G**) survival (*n* = 73). Statistical analysis was performed by multivariate Cox regression analyses. The data are presented as the mean ± SD. **H** The proposed working model of FLI1 in immune suppression and ICB resistance in NPC. Source data are provided as a Source Data file.

A 6-week-old female, SPF humanized NSG mouse model (Shanghai Model Organisms) was created through the administration of $5 \times 10^6$ human peripheral blood mononuclear cells (PBMCs) via tail vein injection. The levels of engraftment were assessed by measuring the proportion of CD45$^+$ human cells in peripheral blood. The mice were subcutaneously inoculated with $1 \times 10^6$ WT or *FLI1*-KO HK1 cells. On the day 6, the mice were randomly assigned to treatment with anti-PD-1 (200 ug/dose, Camrelizumab), YK-4-279 (60 mg/kg), anti-PD-1 plus YK-4-279, or isotype control, and the mice were sacrificed on day 24. The weights of the excised tumors were documented and recorded.

## Cell lines
The human nasopharyngeal epithelial cell line NP69 and NPC cell lines HK1, C666-1 were provided by Professor Musheng Zeng at Sun Yat-sen University Cancer Center (Guangzhou). The human NPC cell lines NPC43, NPC38 and NPC53 were purchased from the NPC Tissue Bank. HEK-293T and MC38 cells were purchased from the American Type Culture Collection (ATCC). NP69 cell line was cultured in K-SFM with rEGF and BPE (Gibco, 17005042). All other cell lines were maintained in RPMI-1640 (Corning, 10-040-CVRC) or DMEM (Corning, 10-013-CVRC) medium, with the addition of 15% fetal bovine serum (FBS; ExCell Bio, FSP500) and 100 U/ml penicillin-streptomycin (Gibico, 15140122). NPC43, NPC38 and NPC53 cells were also supplemented with 4 uM Y-27632 (MedChemExpress, HY-10583).

## Lentivirus-mediated gene transfer
HEK-293T cells were co-transfected with pSPAX2, pMD2G and pSIN-EF2-puro vector coding IDO1, CBP and STAT1. Subsequently, after an eight-hour incubation period post-transfection, the cell culture medium was replaced with Ultra-culture medium (Lonza, 12-725 F). Following a 48-h incubation period, the cell virus supernatant was collected, tittered, and utilized to infect the designated cells overnight. To ascertain successful infection, puromycin (2 µg/ml, Solarbio, P8230) agents were employed.

## Short interfering RNAs transfer
The short interfering RNA (siRNA) molecules were designed (Supplementary Table 2) and synthesized by GenePharma (Shanghai, China). The transfection of siRNA was carried out using Lipofectamine@3000 (Invitrogen, L3000015) following the manufacturer's guidelines and instructions.

## Single-cell isolation and flow cytometry assay
The tumor tissues obtained from C57BL/6 mice were digested into single-cell suspensions by utilizing the mouse Tumor Dissociation Kit (Miltenyi Biotec, 130-096-730) following the manufacturer's guidelines and instructions. Before stained with different antibodies, all samples were stained with Live/Dead dye (Biolegend, Zombie Aqua Fixable Viability Kit, 100×, 423105) for a duration of 15 min at room temperature. For detection of exhausted CD8$^+$ T cell in tumors of animal, antibodies against CD45, CD8, PD-1 and TIM-3 were diluted 1:100 with fluorescence-activated cell sorting buffer, and stained cells. To assess the function of CD8$^+$ T cells, the cells were subjected to stimulation using a cell stimulation cocktail (Invitrogen, 00-4975-03) for a duration of 4 h at 37 °C. Following stimulation, the cells were permeabilized using an intracellular staining kit (Invitrogen, 88-8824-00) and subsequently stained with anti-IFN-γ or anti-TNF antibody to detect the intracellular production of these cytokines. For detection of the proportion of Tregs, cells were stained with CD45, CD4, CD25 and FoxP3 following incubation with FoxP3 Fixation/Permeabilization working solution at 4 °C overnight in the dark (Invitrogen, 00-5523-00). All samples were analyzed using a flow cytometer (cytoFLEX LX, Beckman Coulter) with CytExpert software (Beckman Coulter). The antibodies used are listed in Supplementary Table 3.

## Isolation of T cells and flow cytometry assay
Human CD4$^+$ and CD8$^+$ T cells were isolated from previously frozen peripheral blood mononuclear cells via EasySep™ Human CD4$^+$ (Stem Cell, 17952) and CD8$^+$ T cell isolation kit (Stem Cell, 17953) according to the manufacturer's instructions and cultured in ImmunoCult™ -XF T cell Expansion Medium (Stem Cell, 10981) with IL-2 (10 ng/ml, Pepro-Tech, 200-02-50). To activate T cells, ImmunoCult™-Human CD3/CD28 T Cell Activator (Stem Cell, 10991) was added to the cell suspension. NPC cells were seeded in 12-well plate, and cocultured with activated CD4$^+$ and CD8$^+$ T cells at a ratio of 1:2 for 48 h. CD8$^+$ T cells were collected to stain with CD45, PD-1 and TIM-3, and CD4$^+$ T cells were stained with CD45, CD25 and FoxP3. Samples were analyzed with a flow cytometry. The antibodies used are listed in Supplementary Table 3.

## Cytokines release and Kyn production
Cytokine (IFN-γ and TNF-α) production in the supernatants was quantified by ELISA kits (Biolegend, 430104 and 430204) according to the manufacturer's instructions and Kyn production was measured by ELISA kit (LDN Labor Diagnostika Nord and Immundiagnostik, BA E-2200R) according to the supplier's protocols.

## Western blot analysis
Protein was extracted from the cells using RIPA lysis buffer (Byotime, P0013) supplemented with protease and phosphatase inhibitors (Byotime, P1045), separated by SDS-polyacrylamide gels, and then transferred to NC membranes (Merck Millipore, HATF00010). The membranes were blocked with 5% bovine serum albumin and incubated with primary antibodies. After that, a peroxidase-conjugated secondary antibody was applied to the membranes. The antigen-antibody interaction was visualized using an enhanced chemiluminescence assay (ECL, Thermo, 32106). The antibodies used are listed in Supplementary Table 3, and the unprocessed scans of the Western blots are provided in the Source data.

## RNA extraction and RT-qPCR
RNA extraction was performed with the AllPrep RNA/DNA Mini Kit (Qiagen, 80204) according to manufacturer's instructions. Complimentary DNA (cDNA) was synthesized using random primers and M-MLV reverse transcriptase (Promega, M1705). The qRT-PCR assays were carried out on a CFX96 Touch sequence detection system (Bio-Rad) or a LightCycler 480 System (Roche), with a ChamQ SYBR qPCR Master Mix (Vazyme, Q311-03). The data were standardized to the expression of *GAPDH*. The primer sequences are listed in Supplementary Table 2.

## RNA-Seq analysis
**Library preparation.** Total RNA from *FLI1*-KO and WT HK1 cells was extracted to conduct RNA-seq analysis. The NEBNext® Ultra™ RNA Library Prep Kit for Illumina® (NEB) was employed for generating sequencing libraries, following the manufacturer's guidelines. Index codes were incorporated to label sequences for each sample uniquely. Briefly, mRNA was isolated from the total RNA using poly-T oligo-attached magnetic beads. Fragmentation of the mRNA was performed using divalent cations at an elevated temperature in NEBNext First Strand Synthesis Reaction Buffer (5×). First-strand cDNA was synthesized utilizing random hexamer primers and RNase H. Subsequently, second-strand cDNA synthesis was carried out employing buffer, dNTPs, DNA polymerase I, and RNase H. The resulting library fragments were purified using QiaQuick PCR kit, eluted with EB buffer, and then subjected to terminal repair, A-tailing, and adapter addition. Afterward, the targeted library products were retrieved and subjected to PCR amplification to complete the library preparation. The RNA concentration of the library was measured using the Qubit® RNA Assay Kit on the Qubit® 3.0 system and was subsequently diluted to 1 ng/mL. The insert size of the library was assessed using the Agilent Bioanalyzer 2100 system (Agilent Technologies), and the accurately quantified

insert size with a concentration greater than 10 nM was determined using the StepOnePlus™ Real-Time PCR System. For clustering, the index-coded samples were processed on a cBot cluster generation system using the TruSeq PE Cluster Kit v3-cBot-HS (Illumina) as per the manufacturer's instructions. After cluster generation, the libraries were sequenced on an Illumina Novaseq platform, generating 150 bp paired-end reads.

**Data analysis.** RNA-Seq data underwent adapter trimming, and data quality assessment was performed using fastp software (v 0.23.2) prior to any data filtering steps. Subsequently, the reads were aligned to the human reference genome (GRCh38.p12 assembly) using HISAT2 software (v 2.0.4). The aligned reads were then assembled into transcripts or genes using StringTie software (v 1.3.4d) and the genome annotation file (hg38_ucsc.annotated.gtf). For quantification purposes, the relative abundance of the transcripts/genes was determined using a normalized metric called FPKM (Fragments Per Kilobase of transcript per Million mapped reads).

Statistical analysis was conducted using the R software environment (http://www.r-project.org/). For the differential expression analysis of three groups, the DESeq2 software (v 1.30.1) was utilized. A corrected $P$-value threshold of 0.05 and an absolute fold change of 2 were set as criteria to identify significantly differentially expressed genes. Kyoto Encyclopedia of Genes and Genomes (KEGG) enrichment analysis of differentially expressed genes was implemented by the clusterProfiler (v 4.2.0).

## ChIP-seq analysis

**Library preparation.** HK1 cells were treated with PBS or IFN-γ for 24 h and crosslinked with 1% formaldehyde for 10 min at room temperature, followed by quenching with 125 mM glycine. The chromatin was subsequently fragmented, and the resulting chromatin fragments were pre-cleared before undergoing immunoprecipitation using Protein A + G Magnetic beads coupled with anti-Flag antibodies. Following this, reverse crosslinking was performed, and both the ChIP and input DNA fragments were subjected to end-repair and A-tailing using the NEBNext End Repair/dA-Tailing Module (NEB). Adapter ligation was carried out using the NEBNext Ultra Ligation Module (NEB). The DNA libraries were amplified through 15 cycles of PCR and were then subjected to sequencing using the Illumina Hi-Seq platform with paired-end 2 × 150 as the sequencing mode.

**Data analysis.** Raw reads underwent a filtering process to obtain high-quality clean reads, which involved removing sequencing adapters, short reads (length <35 bp), and low-quality reads using Cutadapt v1.18. The high read quality was further confirmed using FastQC. Next, the clean reads were aligned to the human genome (assembly GRCh38) utilizing the Bowtie2 v2.3.4.1 software. Duplicate reads were subsequently eliminated using picard MarkDuplicates. Peak detection was carried out using the MACS v2.1.2 peak finding algorithm, with a $P$-value cutoff set at 0.01 to identify statistically significant peaks. Annotation of the identified peak sites to gene features was accomplished using the ChIPseeker R package.

## ATAC-seq analysis

**Library preparation.** We harvested 50,000 cells, which were then subjected to centrifugation at $500 \times g$ for 5 min. Afterward, a cold 1 × PBS wash was performed, followed by centrifugation at $500 \times g$ for 5 min. Subsequently, cell lysis occurred using a cold lysis buffer comprising 10 mM Tris-HCl (pH 7.4), 10 mM NaCl, 3 mM MgCl2, and 0.1% IGEPAL CA-630. Nuclei were promptly spun at $500 \times g$ for 10 min at 4 °C, with subsequent discarding of the supernatant. The resulting pellet was immediately resuspended in a transposase reaction mix (25 μL 2× TD buffer, 2.5 μL Transposase (Illumina), and 22.5 μL of nuclease-free water). Transposition was carried out at 37 °C for 30 min.

Following transposition, purification was accomplished using a Qiagen Minelute kit. After purification, library fragments were amplified using 1× NEBnext PCR master mix and 1.25 μM custom Nextera PCR primers. The PCR conditions involved an initial step at 72 °C for 5 min, followed by denaturation at 98 °C for 30 s, and subsequent thermocycling at 98 °C for 10 s, 63 °C for 30 s, and 72 °C for 1 min. A final extension step was performed at 72 °C for 5 min. The libraries were then purified using a Qiagen PCR cleanup kit, resulting in the final library.

**Data analysis.** Raw sequencing reads underwent a series of pre-processing steps to obtain high-quality clean reads. This involved removing sequencing adapters, short reads with a length less than 35 bp, and low-quality reads using Trimmomatic v0.3. Additionally, FastQC was employed to ensure the quality of the resulting reads. The clean reads were then aligned to the human genome (assembly GRCh38) using the Bowtie2 v2.3.4.1 software. Alignments with low mapping quality scores (MAPQ < 30) were filtered out using samtools. To further improve data quality, duplicate reads were removed using the picard MarkDuplicates (http://broadinstitute.github.io/picard). For peak detection, the MACS v2.1.2 peak finding algorithm was utilized with a $q$-value cutoff of 0.05 to identify significant peaks. To evaluate the reproducibility of the high-throughput experiments, the irreproducible discovery rate (IDR v2.0.2) analysis was performed, and only reproducible peaks with an IDR value of 0.05 or less were retained. Next, the peak sites were annotated to gene features using the ChIPseeker R package. To detect potential motifs within the identified peaks, the MEME suite (http://meme-suite.org/) was employed. Lastly, differential peaks between two comparison groups were identified using The DiffBind with a $q$-value cutoff of 0.05.

## ChIP-qPCR

The ChIP assays were performed using the SimpleChIP@ Enzymatic Chromatin IP kit (Cell Signaling Technology). Approximately $1 \times 10^7$ cells were crosslinked with 1% formaldehyde and subsequently quenched with 125 mM glycine to fix the protein-DNA interactions. The cells were then lysed and sonicated to obtain chromatin fragments. The chromatin was immunoprecipitated using specific antibodies (Flag, H3K9Ac, H3K14Ac, H3K18Ac, H3K27Ac, and igG) targeting the proteins of interest. The DNA fragments bound to the immunoprecipitated proteins were then isolated and quantified by qPCR. The primer sequences used in the qPCR are listed in Supplementary Table 2, and the antibodies employed in the immunoprecipitation are listed in Supplementary Table 3.

## Dual-luciferase reporter assay

NPC cells were transfected with Firefly luciferase reporter plasmids along with a control vector containing the Renilla luciferase gene (pRLTK). The Renilla luciferase construct served as an internal control to which the Firefly luciferase values were normalized. After 48 h of transfection, Firefly and Renilla signals were quantified using the Dual Luciferase Reporter Assay Kit (Promega, E1910).

## Metabolite determination

The mouse tumors were pulverized in liquid nitrogen and weighed to obtain the appropriate amount for metabolite extraction. Metabolites were extracted using an ice-cold extraction solvent (80% methanol/water, 1 ml). The tissue was homogenized with a homogenizer to achieve a uniform suspension and incubated on ice for an additional 10 min. The resulting extract was then subjected to centrifugation at $20,000 \times g$ for 10 min at 4 °C to separate the supernatant from the cellular debris. The supernatant was transferred to a new tube, dried using vacuum concentrator, and subsequently redissolved for HPLC/MS analysis. The HPLC/MS analysis was conducted using an Agilent 6495A Triple Quadrupole LC/MS System. The column temperature was maintained at a constant 35 °C, and the mobile phase A and B were

composed of an aqueous solution containing 10 mM ammonium acetate and 0.1% ammonia water, and acetonitrile containing 10 mM ammonium acetate and 0.1% ammonia water, respectively. The HPLC gradient was set as follows: 0 min, 85% B; 1.5 min, 85% B; 5.5 min, 30% B; 8 min, 30% B; 10 min, 85% B; and 12 min, 85% B. The flow rate was 0.3 ml/min, and 1 μl of the sample was injected for HPLC-MS analysis. The data obtained were processed using Agilent MassHunter Quantitative Analysis to quantify the metabolites. The retention times and mass fragmentation signatures of all metabolites were validated using pure standards.

### IHC staining and scoring
Paraffin-embedded tissue samples were cut into sections of 3μm thickness. The specimens were incubated with primary antibodies against FLI1 (1:200), IDO1 (1:200), cleaved-Caspase-3 (1:200), and Ki-67 (1:200) overnight at 4 °C. Immunodetection was performed the following day using DAB (Dako, K5007) following the manufacturer's instructions. Images of the stained samples were captured using an AxioVision Rel.4.6 computerized image analysis system (Carl Zeiss). IHC staining of FLI1 and IDO1 were scored independently by two experienced pathologists using the IRS system to ensure consistency and accuracy. The intensity of staining was categorized into four scores: 0 (no brown particle staining), 1 (light brown particles), 2 (moderate brown particles), and 3 (dark brown particles). The percentage of positive tumor cells was classified into four scores: 1 (<10% positive cells), 2 (10–40% positive cells), 3 (40–70% positive cells), and 4 (>70% positive cells). The IRS was determined by multiplying the two scores together, resulting in a range of scores from 0 to 12. High levels of expression were defined as an IRS score of ≥6, whereas low levels were defined as an IRS score of <6. The percentages of cleaved Caspase-3 and Ki-67 positive cells were performed using HALO version 3.2.1851 (Indica Labs)[41]. The specific antibodies used in the IHC analysis are listed in Supplementary Table 3.

### Bioinformatics analysis
We obtained RNA-seq data for various types of tumors from TCGA and GEO database (GEO102349), and generating a score that reflects the T-cell exhaustion status for each sample by ssGSEA. People were divided into quantiles on the basis of T cell exhaustion signatures (*PD-1*, *TIM-3*, *TIGIT*, *LAG-3*, *CXCL13* and *LAYN*) through ssGSEA by R software environment (http://www.r-project.org/). Differentially expressed genes between tumors with high (top 25%) and low (bottom 25%) T cell exhaustion signatures were identified (log2FC ≥ 0.3, adjust $P$ value < 0.05). Venn diagram was used to show the overlap of differentially expressed genes of these two groups in NPC, COAD, READ, LUSC and SKCM and a core transcription factor gene set (https://jaspar.elixir.no/). RNA-seq data from tumors with high (top 25%) and low (bottom 25%) *FLI1* expression were used for GSEA (version 4.3.2) analysis. We constructed a Kyn metabolism signature using Kyn synthesis-related genes (*TDO2*, *IDO1*, *IDO2*, *AFMID*, *KMO*, *KYNU*, *KYAT3*, *KYAT1* and *AADAT*) through ssGSEA by R software environment, and analyzed the association between Kyn metabolism and T cell exhaustion in NPC. In addition, we constructed a regulatory T cell signature (*FOXP3*, *CTLA4*, *CCR8*, *TNFRSF9* and *IL2RA*) and analyzed the association between Kyn metabolism and Treg gene signatures.

### Quantification and statistical analysis
All experiments were repeated at least three times to ensure reproducibility. The results are presented as the mean ± SD as indicated and were subjected to statistical analysis using two-tailed Student's $t$ test, two-way ANOVA or one-way ANOVA with the Tukey's multiple comparisons test, as appropriate. A $P$ value of less than 0.05 was considered statistically significant. The chi-square ($\chi^2$) test was used to compare clinical characteristics. Survival curves were generated using the Kaplan–Meier method, and differences among groups were

compared using the log-rank test to assess the significance of the survival differences. To identify independent prognostic factors, multivariate analysis was performed using a Cox proportional hazards regression model. All statistical analyses were conducted using the SPSS version 27.0 statistical software, and GraphPad Prism version 9.0 software was also used for data analysis and visualization.

### Reporting summary
Further information on research design is available in the Nature Portfolio Reporting Summary linked to this article.

## Data availability
The raw sequence data of RNA-seq, ChIP-seq and ATAC-seq data generated in this study have been deposited in the GEO database under accession code GSE247898. TCGA datasets for COAD, READ, LUSC and SKCM, OV, STAD and BLCA were all obtained from UCSC Xena [https://xenabrowser.net/datapages/]. The NPC sequencing data used in this study are available in the GEO database under accession code GSE102349. The core transcription factor gene set was obtained from the JASPAR database (https://jaspar.elixir.no/). The remaining data generated in this study are provided in the Supplementary Information/Source Data file. Source data are provided with this paper.

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

## Acknowledgements

This research was supported by National Natural Science Foundation of China (82172949, 82372650, 81972569 to W.G.D.) and the Guangdong Basic and Applied Basic Research Foundation (2022A1515012508 to W.G.D.). Figures 1F, 6B, 7H, Supplementary Fig. S6B were created with BioRender.com.

## Author contributions

E.N.C., W.G.D., F.Y.X., and M.C. conceived the ideas and designed the experiments. E.N.C., J.W.W., J.J.H., W.C.Z., and H.H.S. performed the experiments. X.N.W., and D.G.L. performed bioinformatics data analysis. E.N.C., X.D.L., and Z.Q.L. performed clinical data analysis. E.N.C., J.W.W., J.J.H., M.C., and D.B.S. analyzed the data. E.N.C., W.G.D., and F.Y.X. wrote the paper. J.W.W., and J.S.H. jointly supervised this work. All authors reviewed and discussed the final version of the paper.

## Competing interests

The authors declare no competing interests.
