## [Peer Review File · Nature Communications]

REVIEWER COMMENTS

Reviewer #1 (Remarks to the Author):

Chen et al reported that FLI1 promotes IFN- γ -induced kynurenine 1 production to impair anti-tumor immunity. The authors found that tumor-intrinsic FLI1 could be a key mediator in impairing T cell anti-tumor immunity through bioinformatic analysis. The authors further found that FLI1 promotes CD8+ T cell exhaustion by enhancing the IFN- γ -IDO1-Kyn axis to mediate T cell immunity suppression. Moreover, FLI1 modulates IDO1 expression at both epigenetic and transcriptional levels by regulating CBP and STAT transcription. Furthermore, a small molecular inhibitor YK-4-279 targeting FLI1 transcriptional activity relieves Kyn-mediated immune suppression, and combined YK-4-279 and anti-PD-1 treatment enhance T cell-mediated anti-tumor immunity. This work is interesting, however, there are some serious concerns that the authors should address before publication.

Comments:

1. Three NPC cell lines used in this study (HONE1, SUNE1, 5-8F, ref to <https://www.cellosaurus.org/>) are contaminated cell lines, thus the experiments conducted in these cell lines could not represent the actual roles of FLI1 in NPC. Furthermore, NPC is an EBV-associated cancer. EBV-positive cell lines (C666-1 or NPC43) should be included in the study.
2. Mouse cell line MC38 is a commonly used murine model for colorectal carcinoma, not NPC type, although it is sensitive to immune checkpoint immunotherapy and endogenous CD8+ T cell responses against neoantigens.
3. In addition to mRNA expression, the authors should examine the protein expression of FLI1 in NPC tumor cells and nasopharyngeal epithelial cells, as it is a transcription factor.
4. Using an in vivo T cell CRISPR screening platform, a recent study has identified that Fli1 is a factor modulating CD8+ T cell effector differentiation with limited effects on memory or exhaustion (Cell. 2021 Mar 4;184(5):1262-1280.e22.). However, in this study, the authors claimed that tumor-intrinsic FLI1 deficiency is a critical determinant in promoting CD8+ T cell exhaustion. The authors should discuss them.

Overall, the current experiments conducted could not delineate FLI1-mediated Kyn metabolism as a novel immune evasion mechanism in NPC.

Reviewer #2 (Remarks to the Author):

The manuscript by Chen et al., investigated a novel role for FLI1 as a therapeutic target for cancer immunotherapy in nasopharyngeal carcinoma (NPC). Authors identify FLI1 as a driver for CD8+ T cell exhaustion and Treg polarization. Mediated through the IDO pathway. Mechanistically, FLI1 regulates the epigenetic modifier CBP, which promotes chromatin accessibility at the IDO promoter region, and signaling through STAT1. Genetic knockdown of FLI1 as well as targeting FLI1 using the small molecule drug YK-4-279 impaired IDO1/Kyn synthesis, thereby abrogating the immunosuppressive functions of FLI1 in NPC. The manuscript is clearly written and data are convincingly presented, and may point at a novel strategy for cancer immunotherapy. There are some concerns and suggestions that in my opinion can strengthen the study:

- Regarding the tumor-intrinsic effect of FLI1 in promoting CD8+ T cell exhaustion and Treg polarization, WT or FLI1-deficient MC38 were transplanted into mice. It was shown that tumor outgrowth of FLI1-deficient tumors was significantly reduced. Though this was paralleled by reduced CD8+ T cell exhaustion (figure 1) and Treg polarization (figure 2), it was also found that FLI1 inhibition reduces MC38 proliferation directly. One might suggest that the reduced tumor outgrowth may be a direct anti-proliferative effect of FLI1-knockdown in tumor cells. The study would therefore be strengthened if authors can show the relative contribution of CD8+ T cells exhaustion and Treg polarization on the delayed tumor outgrowth upon FLI1 knockout by depleting CD8+ T cells or Tregs.

- Authors indicate the FLI1 blockade may be used as a complementary immunotherapeutic strategy next to anti-PD1 treatment. As IFN γ is also known to increase PD-L1 expression via STAT1 on various tumor types, how does FLI1-deficiency affect PD-L1 expression on NPC? And can overexpression of IDO1 in FLI1-deficient MC38 cells impair CD8+ T cell exhaustion in vivo, as was also shown for Treg polarization (figure 2N) Is there an association between FLI1 expression and PD-L1 expression on tumor cells? And how does FLI1 expression compare to PD-L1 expression as a predictor for survival or responsiveness to anti-PD1 treatment? It would be interesting to see if a similar association between patient survival can be observed when patients are stratified based on PDL1 expression by the tumor.

- The authors used the MC38 (i.e. a colon carcinoma) model to identify the role of FLI1 in mediating immune evasion by tumor, but only used NPC lines for in vitro investigation. Authors need to show that FLI1-deficiency also impacts on anti-tumor immunity in in vivo NPC models to substantiate the findings. Especially, since authors conclude 'FLI1-mediated Kyn metabolism as a novel immune evasion mechanism in NPC' (abstract lines 47-48).

- Since FLI1 was identified as a driver of T cell exhaustion in multiple tumor types (figure 1A), the study would significantly be strengthened if authors can show an association between FLI1 expression and survival status in patients with other tumor types, such as colon carcinoma or melanoma.

Reviewer #3 (Remarks to the Author):

Enni Chen and Jiawei Wu show that FLI1 associates with T cell exhaustion. FLI1 regulates the expression of CBP and STAT1, thereby promoting chromatin accessibility and transcription of IDO1 in response to IFN γ , which leads to CD8+ T cell exhaustion and regulatory T cell (Treg) differentiation. Pharmacological inhibition of FLI1 by YK-4-279 inhibits the CBP/STAT1-IDO1-Kyn axis and reduces tumor growth both alone and in combination with ICB. FLI1-IDO1 levels and clinical outcome correlate in patients with nasopharyngeal carcinoma.

This is an interesting and well-performed study, that may have clinical implications.

I only have minor requests:

Your in vivo models are not NPC, maybe put less emphasis on NPC in the summary and introduction.

Please perform at least three independent western blots and show the quantification in a graph throughout the manuscript.

The RNASeq data needs to be made available to the reviewers and later to the public.

Fig.1 Please explain HK1 and 5-8F in the legend

2C the number for p and R are too small

S2 Please explain what the Kyn signatures are and how they were derived in more detail

Legends of Figures 3 and 4: Instead of just writing fold enrichment (relative), please specify of what. Then one can understand the figure without even having to consult the legend.

4G Please show the effects of modulation of FLI1 and CBP here.

Please introduce YK-4-279 better, e.g. the small molecule inhibitor

IDO1 regulation by IFN γ is well established and can be mentioned only very concisely

5C, D you already established that IFN γ is necessary, no need to show cells not treated with IFN γ here, it just makes the graphs busier

I can follow the effects of YK-4-279 on IDO and Kyn, but I wonder whether the effects on STAT1 observed for FLI1 would also negatively affect tumor characteristics, i.e. are IFN γ effects that are positive for tumor patients reduced?

MC38 tumors respond very well to immunotherapy and even though interventions have been successful here they didn't always lead to successful therapies. Can these exciting results be confirmed in an additional model?

The font of the words in the graphical abstract is too small.

If one looks at these data, it is surprising that IDO1 inhibitors failed in clinical trials, this should be discussed including possible compensatory mechanisms such as e.g. IL4I1, which also activates the AHR and is regulated in a similar manner to IDO1.

Our point-by-point responses to the reviewer' comments are below.

Reviewer #1 (Remarks to the Author):

Chen et al reported that FLI1 promotes IFN- γ -induced kynurenine 1 production to impair anti-tumor immunity. The authors found that tumor-intrinsic FLI1 could be a key mediator in impairing T cell anti-tumor immunity through bioinformatic analysis. The authors further found that FLI1 promotes CD8+ T cell exhaustion by enhancing the IFN- γ -IDO1-Kyn axis to mediate T cell immunity suppression. Moreover, FLI1 modulates IDO1 expression at both epigenetic and transcriptional levels by regulating CBP and STAT transcription. Furthermore, a small molecular inhibitor YK-4-279 targeting FLI1 transcriptional activity relieves Kyn-mediated immune suppression, and combined YK-4-279 and anti-PD-1 treatment enhance T cell-mediated anti-tumor immunity. This work is interesting, however, there are some serious concerns that the authors should address before publication.

Response to R1:

We thank the reviewer for the positive comments and nice suggestions. We have followed up with more analyses as you recommended.

1) Three NPC cell lines used in this study (HONE1, SUNE1, 5-8F, ref to <https://www.cellosaurus.org/>) are contaminated cell lines, thus the experiments conducted in these cell lines could not represent the actual roles of FLI1 in NPC. Furthermore, NPC is an EBV-associated cancer. EBV-positive cell lines (C666-1 or NPC43) should be included in the study.

Response:

We thank the reviewer for the insightful comments regarding the use of cell lines in our study. Upon reviewing the Cellosaurus database, we acknowledge that the NPC cell lines HONE1, SUNE1, and 5-8F have been reported as contaminated. We appreciate you bringing this critical issue to our attention, as it is essential to ensure the validity and reliability of our experimental results. To address this, we retest the FLI1 expression between normal nasopharyngeal epithelial cell and uncontaminated NPC cell lines, and replicate our functional and mechanical experiments using NPC43 cell line. The latest results also demonstrate an immunosuppressive role of FLI1 and the existence of FLI1-CBP/STAT1-IDO1-Kyn axis. **(Data have been presented in the revised manuscript).**

2) Mouse cell line MC38 is a commonly used murine model for colorectal carcinoma, not NPC type, although it is sensitive to immune checkpoint immunotherapy and endogenous CD8⁺ T cell responses against neoantigens.

Response:

We thank the reviewer for pointing out the issue regarding the use of in-vivo model. We understand the importance of using an appropriate model system to study the role of FLI1 in NPC. To address this, we have established NPC xenografts using FLI1-knockout (KO) and wild-type (WT) HK1 cells in humanized NOD/SCID/IL2 γ null mice. This model allows us to closely examine the impact of FLI1 on the tumor microenvironment within the context of NPC. Our findings indicate that FLI1 deficiency leads to a reduction in tumor growth (Figure R1. 1A), tumor weight (Figure R1. 1B) and intratumoral kynurenine (Kyn) levels (Figure R1. 1C). Furthermore, we observe that FLI1 deficiency impedes CD8⁺ T cell exhaustion (Figures R1. 1D-1G) and Treg differentiation (Figure R1. 1H), supporting the hypothesis that FLI1 plays a significant role in modulating the immune landscape of NPC. More importantly, our data suggest that combining FLI1 inhibition with anti-PD-1 therapy can synergistically enhance T cell-mediated anti-tumor immunity, resulting in more effective suppression of tumor growth (Figures R1. 1A-1H). These findings underscore the potential therapeutic benefits of targeting FLI1 in combination with immune checkpoint inhibitors to potentiate anti-tumor immune responses in NPC. **The relevant data has been put in the revised manuscript (On page 9, line 255 to page 10, line 265).**

Figure R1 (1A and 1B) Analysis of the tumor growth rate **(1A)** and the endpoint tumor weight **(1B)**. **(1C)** The level of Kyn in tumors was assessed by HPLC-MS. **(1D-1G)** The expression of PD-1 **(1D)** and TIM-3 **(1E)** on CD8⁺ T cells, and the expression of IFN-γ **(1F)** and TNF-α **(1G)** in CD8⁺ T cells isolated from the indicated tumors were measured by flow cytometry. **(1H)** The proportion of CD25⁺ FoxP3⁺ cells among CD4⁺ T cells isolated from the indicated tumors were measured by flow cytometry.

3) In addition to mRNA expression, the authors should examine the protein expression of FLI1 in NPC tumor cells and nasopharyngeal epithelial cells, as it is a transcription factor.

Response:

We are grateful for your suggestion to examine the protein expression levels of FLI1 for providing a more comprehensive understanding of its role in NPC. The results from the Western blot analysis are in alignment with our qPCR findings (**Original Figures S1A and S1B**), showing that FLI1 protein levels are indeed significantly elevated in multiple NPC cell lines when compared to nasopharyngeal epithelial cell line NP69 (**Figure R1.2A**). This enhancement of FLI1 expression is also observed in NPC tissues (**Figure R1.2B**). This further substantiates our initial findings and reinforces the

potential significance of FLI1 as a key transcription factor in NPC. **The relevant data has been put in the revised manuscript (On page 4, line 103).**

Figure R1 (2A) FLI1 protein level in normal nasopharyngeal epithelial and NPC cell lines. **(2B)** FLI1 protein level in normal nasopharynx tissues and NPC tissues.

4) Using an in vivo T cell CRISPR screening platform, a recent study has identified that Fli1 is a factor modulating CD8⁺ T cell effector differentiation with limited effects on memory or exhaustion (Cell. 2021 Mar 4;184(5):1262-1280.e22.). However, in this study, the authors claimed that tumor-intrinsic FLI1 deficiency is a critical determinant in promoting CD8⁺ T cell exhaustion. The authors should discuss them.

Response:

We thank the reviewer for directing our attention to the recent study utilizing an in vivo T cell CRISPR screening platform, which identified FLI1 as a modulator of CD8⁺ T cell effector differentiation with limited impact on memory or exhaustion (PMID: 33636129). We have thoroughly examined this publication and included a discussion of its findings in our revised manuscript (see page 11, line 308 to line 315). We agree that the role of FLI1 in T cell biology is complex and context-dependent. It is important to note that the Kyn synthesis-related genes, such as IDO1, IDO2, and TDO2, are minimally expressed in T cells. This underpins our hypothesis that the absence of a FLI1-IDO1 pathway within T cells may contribute to the limited effects of FLI1 on CD8⁺ T cell exhaustion observed in the study you referenced. Furthermore, we propose that the discrepancy between our findings and this cited study could be attributed to tissue heterogeneity and the multifaceted nature of functions of FLI1. While FLI1 may indeed have a nuanced role within T cells, its impact on the TME via the tumor-intrinsic FLI1-IDO1 axis should not be underestimated. In our study, we provide evidence that tumor cells, as a major component of the TME, can exploit the FLI1-IDO1 axis to

produce Kyn and effectively suppress T cell-mediated immunity. This suggests that tumor-intrinsic FLI1 deficiency could disrupt this immunosuppressive mechanism.

Reviewer #2 (Remarks to the Author)

The manuscript by Chen et al., investigated a novel role for FLI1 as a therapeutic target for cancer immunotherapy in nasopharyngeal carcinoma (NPC). Authors identify FLI1 as a driver for CD8+ T cell exhaustion and Treg polarization. Mediated through the IDO pathway. Mechanistically, FLI1 regulates the epigenetic modifier CBP, which promotes chromatin accessibility at the IDO promoter region, and signaling through STAT1. Genetic knockdown of FLI1 as well as targeting FLI1 using the small molecule drug YK-4-279 impaired IDO1/Kyn synthesis, thereby abrogating the immunosuppressive functions of FLI1 in NPC. The manuscript is clearly written and data are convincingly presented, and may point at a novel strategy for cancer immunotherapy. There are some concerns and suggestions that in my opinion can strengthen the study:

Response to R2:

We thank the reviewer for the positive comments and nice suggestions. We have followed up with more analyses as you recommended.

1) Regarding the tumor-intrinsic effect of FLI1 in promoting CD8+ T cell exhaustion and Treg polarization, WT or FLI1-deficient MC38 were transplanted into mice. It was shown that tumor outgrowth of FLI1-deficient tumors was significantly reduced. Though this was paralleled by reduced CD8+ T cell exhaustion (figure 1) and Treg polarization (figure 2), it was also found that FLI1 inhibition reduces MC38 proliferation directly. One might suggest that the reduced tumor outgrowth may be a direct anti-proliferative effect of FLI1-knockdown in tumor cells. The study would therefore be strengthened if authors can show the relative contribution of CD8+ T cells exhaustion and Treg polarization on the delayed tumor outgrowth upon FLI1 knockout by depleting CD8+ T cells or Tregs.

Response:

We thank the reviewer for the nice comments. The reviewer raises legitimate concerns on whether reduced tumor outgrowth might be a direct anti-proliferative effect of FLI1-knockdown in tumor cells. To eliminate this potential distraction, we have conducted

additional experiments using depleting antibodies against CD8⁺ and CD4⁺ T cells to discern the specific contributions of immune modulation to tumor outgrowth. Our results indicate that the depletion of these T cell populations significantly mitigates the tumor-suppressive effects observed with FLI1 deficiency (Figure. R2.1A and R2.1B). Moreover, IHC examination of mouse tumors reveals no alteration in Ki-67 levels in FLI1-KO tumors relative to FLI1-WT counterparts (Original Figure 1O, Figure S1H). These findings strongly suggest that the impaired tumor growth in FLI1-deficient tumors is critically dependent on the presence of an active immune response, thereby underscoring the importance of FLI1 in creating an immunosuppressive environment that facilitates tumor progression. The relevant data has been put in the revised manuscript (On page 6, line 164 to line 170).

Figure R2 (1A and 1B) WT or FLI1-KO MC38 cells were inoculated into C57BL/6 mice. The depleting antibodies against CD8⁺ and CD4⁺ T cells were administered intraperitoneally every 3 days. The tumor growth rate (1A) and endpoint tumor weight (1B) are reported.

2) Authors indicate the FLI1 blockade may be used as a complementary immunotherapeutic strategy next to anti-PD1 treatment. As IFN γ is also known to increase PD-L1 expression via STAT1 on various tumor types, how does FLI1-deficiency affect PD-L1 expression on NPC? And can overexpression of IDO1 in FLI1-deficient MC38 cells impair CD8⁺ T cell exhaustion *in vivo*, as was also shown for Treg polarization (figure 2N) Is there an association between FLI1 expression and PD-L1 expression on tumor cells? And how does FLI1 expression compare to PD-L1 expression as a predictor for survival or responsiveness to anti-PD1 treatment? It would be interesting to see if a similar association between patient survival can be observed when patients are stratified based on PDL1 expression by the tumor.

Response:

We thank the reviewer for the valuable comments. The reviewer raises an interesting question on whether IFN- γ -induced PD-L1 expression is regulated by FLI1. To address it, we applied a qPCR assay and find that compared with that of IDO1, FLI1 deficiency takes modest effects in reduction of PD-L1 transcription in NPC cell lines in the presence of IFN- γ in vitro. These results reflect the importance of FLI1-mediated epigenetic regulation for IDO1 expression (Figure R2.2A). Of note, we perform an additional qPCR assay using humanized mouse tumors, and observe that FLI1 KO results in no significant alteration in PD-L1 transcription (Figure R2.2B). This phenomenon can be explained by the fact that the TME provides a further enhanced niche for cancer immune escape by augmenting PD-L1 expression induced by other cytokines, such as TNF- α and IL-6 (PMID: 31668929).

Figure R2 (2A) PD-L1 mRNA level in FLI1-KO and WT HK1 upon IFN- γ stimulation. **(2B)** PD-L1 mRNA level in FLI1-KO and WT groups of humanized NSG mice.

Regarding the clinical significance of FLI1 and PD-L1 expression, survival analyses utilizing the TCGA database reveal that elevated FLI1 expression correlates with diminished overall survival across multiple cancer types (Figure R2.2C). However, PD-L1 expression does not demonstrate a significant association with overall survival (Figure R2.2D). Furthermore, due to the lack of NPC database containing anti-PD-1 therapy data, we use database from several other tumor types, such melanoma and urothelium carcinoma, to assess the predictive value of FLI1 expression on the response to anti-PD-1 therapy. Our analysis yielded no significant correlation between FLI1 expression and the efficacy of anti-PD-1 treatment (data unrepresented). Thus, we do not compare their values in predicting overall survival nor response to anti-PD-1 therapy.

R2.2C

R2.2D

Figure R2 (2C and 2D) Kaplan-Meier survival analysis of the association between overall survival and the expression levels of FLI1 (**2C**) and PD-L1 (**2D**) across patient cohorts with colon adenocarcinoma (COAD), rectal adenocarcinoma (READ), lung squamous cell carcinoma (LUSC), and bladder urothelial carcinoma (BLCA) as sourced from the TCGA database.

In response to the reviewer's concern on the potential impact of IDO1 overexpression in FLI1-deficient MC38 cells on T-cell-mediated antitumor immunity *in vivo*, we have conducted supplementary IDO1 rescue experiments (**Figure R2.2E-2L**), further confirming the role of FLI1-IDO1 axis in TME remodeling. **The relevant data has been put in the revised manuscript (On page 6, line 153 to line 156 and line 162 to line 164).**

Figure R2 WT, FLI1-KO or FLI1-KO + IDO1-overexpression (OE) MC38 cells were inoculated into C57BL/6 mice, and tumors were dissected at day 21. **(2E)** The level of Kyn in tumors was assessed by HPLC-MS. **(2F and 2G)** The tumor growth rate **(2F)** and endpoint tumor weight **(2G)** are reported. **(2H-2K)** The expression of PD-1 **(2H)** and TIM-3 **(2I)** on CD8⁺ T cells, and the expression of IFN- γ **(2J)** and TNF- α **(2K)** in CD8⁺ T cells isolated from the indicated tumors were measured by flow cytometry. **(2L)** The proportion of CD25⁺ FoxP3⁺ cells among CD4⁺ T cells isolated from the indicated tumors were determined by flow cytometry.

3) The authors used the MC38 (i.e. a colon carcinoma) model to identify the role of FLI1 in mediating immune evasion by tumor, but only used NPC lines for in vitro investigation. Authors need to show that FLI1-deficiency also impacts on anti-tumor immunity in in vivo NPC models to substantiate the findings. Especially, since authors conclude ‘FLI1-mediated Kyn metabolism as a novel immune evasion mechanism in NPC’ (abstract lines 47-48).

Response:

We thank the reviewer for pointing out the issue regarding the use of in-vivo model. We understand the importance of using an appropriate model system to study the role of FLI1 in NPC. To address this, we have established NPC xenografts using FLI1- KO and WT HK1 cells in humanized NOD/SCID/IL2 γ null mice. This model allows us to closely examine the impact of FLI1 on the tumor microenvironment within the context of NPC. Our findings indicate that FLI1 deficiency leads to a reduction in tumor growth **(Figure R2. 3A)**, tumor weight **(Figure R2. 3B)** and intratumoral Kyn levels **(Figure R2. 3C)**. Furthermore, we observe that FLI1 deficiency impedes CD8⁺ T cell exhaustion **(Figures R2. 3D-3G)** and Treg differentiation **(Figure R2. 3H)**, supporting the hypothesis that FLI1 plays a significant role in modulating the immune landscape of NPC. More importantly, our data suggest that combining FLI1 inhibition with anti-PD-1 therapy can synergistically enhance T cell-mediated anti-tumor immunity, resulting in more effective suppression of tumor growth **(Figures R2. 3A-3H)**. These findings underscore the potential therapeutic benefits of targeting FLI1 in combination with immune checkpoint inhibitors to potentiate anti-tumor immune responses in NPC. **The relevant data has been put in the revised manuscript (On page 9, line 255 to page 10, line 265).**

Figure R2 (3A and 3B) Analysis of the tumor growth rate (3A) and the endpoint tumor weight (3B). (3C) The level of Kyn in tumors was assessed by HPLC-MS. (3D-3G) The expression of PD-1 (3D) and TIM-3 (3E) on CD8⁺ T cells, and the expression of IFN-γ (3F) and TNF-α (3G) in CD8⁺ T cells isolated from the indicated tumors were measured by flow cytometry. (3H) The proportion of CD25⁺ FoxP3⁺ cells among CD4⁺ T cells isolated from the indicated tumors were measured by flow cytometry.

4) Since FLI1 was identified as a driver of T cell exhaustion in multiple tumor types (figure 1A), the study would significantly be strengthened if authors can show an association between FLI1 expression and survival status in patients with other tumor types, such as colon carcinoma or melanoma.

Response:

We thank the reviewer for the valuable suggestion to explore the association between FLI1 expression and survival outcomes across various tumor types. We agree that such analysis would indeed fortify the clinical implications of our findings. We add the prognostic analysis of FLI1 in multiple tumor types (Figure R2.4). The relevant data has been put in the revised manuscript (On page 4, line 104 to line 107).

Figure R2.4

Figure R2.4 Kaplan-Meier survival analysis of the association between overall survival and the expression levels of FLI1 and PD-L1 across patient cohorts with COAD, READ, LUSC, BLCA, ovarian cancer (OV) and stomach cancer (STAD) as sourced from the TCGA database.

Reviewer #3 (Remarks to the Author):

Enni Chen and Jiawei Wu show that FLI1 associates with T cell exhaustion. FLI1 regulates the expression of CBP and STAT1, thereby promoting chromatin accessibility and transcription of IDO1 in response to IFN γ , which leads to CD8+ T cell exhaustion and regulatory T cell (Treg) differentiation. Pharmacological inhibition of FLI1 by YK-4-279 inhibits the CBP/STAT1-IDO1-Kyn axis and reduces tumor growth both alone and in combination with ICB. FLI1-IDO1 levels and clinical outcome correlate in patients with nasopharyngeal carcinoma. This is an interesting and well-performed study, that may have clinical implications.

Response to R3:

We thank the reviewer for the positive comments and valuable suggestions. We have followed up with more analyses as you recommended.

1) Your in vivo models are not NPC, maybe put less emphasis on NPC in the summary and introduction.

Response:

We thank the reviewer for the comments about the emphasis on the tumor type in our study. To emphasize the immunomodulatory role of FLI1 in NPC, we have established NPC xenografts using FLI1- KO and WT HK1 cells in humanized NOD/SCID/IL2ry null mice. This model allows us to closely examine the impact of FLI1 on the tumor

microenvironment within the context of NPC. Our findings indicate that FLI1 deficiency leads to a reduction in tumor growth (Figure R3.1A), tumor weight (Figure R3.1B) and intratumoral Kyn levels (Figure R3.1C). Furthermore, we observe that FLI1 deficiency impedes CD8⁺ T cell exhaustion (Figures R3.1D-1G) and Treg differentiation (Figure R3.1H), supporting the hypothesis that FLI1 plays a significant role in modulating the immune landscape of NPC. More importantly, our data suggest that combining FLI1 inhibition with anti-PD-1 therapy can synergistically enhance T cell-mediated anti-tumor immunity, resulting in more effective suppression of tumor growth (Figures R3.1A-1H). These findings underscore the potential therapeutic benefits of targeting FLI1 in combination with immune checkpoint inhibitors to potentiate anti-tumor immune responses in NPC. **The relevant data has been put in the revised manuscript (On page 9, line 255 to page 10, line 265).**

Figure R3 (1A and 1B) Analysis of the tumor growth rate (1A) and the endpoint tumor weight (1B). (1C) The level of Kyn in tumors was assessed by HPLC-MS. (1D-1G) The expression of PD-1 (1D) and TIM-3 (1E) on CD8⁺ T cells, and the expression of IFN-γ (1F) and TNF-α (1G) in CD8⁺ T cells isolated from the indicated tumors were measured by flow cytometry. (1H) The proportion of CD25⁺ FoxP3⁺ cells among CD4⁺ T cells isolated from the indicated tumors were measured by flow cytometry.

2) Please perform at least three independent western blots and show the quantification in a graph throughout the manuscript.

Response:

We appreciate the reviewer's suggestion. We have presented quantification data of three independent western blot in supplementary figures.

3) The RNASeq data needs to be made available to the reviewers and later to the public.

Response:

We thank the reviewer for the nice suggestion. We have uploaded RNASeq data to GEO database (GSE247898).

4) Fig.1 Please explain HK1 and 5-8F in the legend

Response:

We appreciate the reviewer's suggestion. We have concretely described the used cell models in revised manuscript.

5) 2C the number for p and R are too small

Response:

We thank the reviewer for the kind suggestion. We have enlarged the font of p and R values

6) S2 Please explain what the Kyn signatures are and how they were derived in more detail

Response:

We appreciate the reviewer's suggestion. We use the Kyn synthesis-related genes presented in a previous study (PMID: 30760888) to construct this signature. The more specific description has been added in Methods of revised manuscript (see page 23, line 652 to line 655).

7) Legends of Figures 3 and 4: Instead of just writing fold enrichment (relative), please specify of what. Then one can understand the figure without even having to consult the legend.

Response:

We thank the reviewer for the kind suggestion. For the convenience of the reader to understand the figures, we have corrected Y axis titles in corresponding graphs.

8) 4G Please show the effects of modulation of FLI1 and CBP here.

Response:

We appreciate the reviewer's suggestion. We have added ATAC-seq data in the case of FLI1 KO in the graph file (Figure R3.2). Due to completion of core mechanism study and fund saving, we do not further conduct ATAC-seq in the case of CBP KO.

Figure R3.2

Figure R3.2 The chromatin accessibility at the STAT1 gene locus in FLI1-KO and WT HK1 cells was measured by ATAC-seq following IFN- γ treatment for 24 hours.

9) Please introduce YK-4-279 better, e.g. the small molecule inhibitor

Response:

We thank the reviewer for the nice suggestion. We have re-written this part in the revised manuscript (see page 13, line 354 to line 360).

10) IDO1 regulation by IFN γ is well established and can be mentioned only very concisely

Response:

We appreciate the reviewer's suggestion. We have deleted original description. Instead, we introduce the current finding explaining IDO1 upregulation in tumors (see page 12, line 322 to line 329).

11) 5C, D you already established that IFN γ is necessary, no need to show cells not treated with IFN γ here, it just makes the graphs busier

Response:

We thank the reviewer for the nice comments. We have readjusted and presented these results in revised graph file.

12) I can follow the effects of YK-4-279 on IDO and Kyn, but I wonder whether the effects on STAT1 observed for FLI1 would also negatively affect tumor characteristics, i.e. are IFN γ effects that are positive for tumor patients reduced?

Response:

We thank the reviewer for the insightful query regarding the interplay between FLI1 deficiency and the beneficial effects of IFN- γ signaling in tumor suppression mediated through STAT1. IFN- γ is well-known for its tumor-suppressing activities, which include enhancement of antigen presentation by tumor cells and the recruitment of T cells, primarily through the activation of STAT1, which subsequently upregulates MHC-I (HLA-A/B/C) and CXCL9/10/11 expression. Our in vitro qPCR assays have indicated a slight reduction in MHC-I (Figure R3.3A) and CXCL9/10/11 mRNA levels (Figure R3.3B) following FLI1 KO. Additionally, we have analyzed mRNA extracted from humanized mouse tumor samples to assess the transcriptional levels of these genes. Interestingly, we do not observe a significant change in the mRNA levels of most genes under investigation (Figures R3.3C and 3D). We consider it is reasonable to observe this phenomenon. We observed that while FLI1 deficiency does lead to a partial decrease in phosphorylated STAT1 (the activated form) levels in NPC cell lines under in-vitro conditions, it does not completely abrogate the activation of STAT1. Furthermore, it is reported that within TME, other inflammatory cytokines may activate alternative pathways of tumor cells, such as the NF- κ B pathway, which also play a role in promoting MHC-I and CXCL9/10/11 transcription, thereby potentially compensating for reduced STAT1 activation. In light of these findings, we propose that the partial reduction in IFN- γ -mediated anti-tumor effects due to FLI1 deficiency is likely mitigated by other stimulatory mechanisms within the TME. Therefore, we posit that the therapeutic benefits gained from inhibiting FLI1 activity could surpass potential disadvantages associated with partly diminished STAT1 activation.

Figure R3 (3A) CXCL9, CXCL10 and CXCL11 mRNA levels in FLI1-KO and WT HK1 upon IFN-γ stimulation. **(3B)** HLA-A, HLA-B and HLA-C mRNA levels in FLI1-KO and WT HK1 upon IFN-γ stimulation. **(3C)** CXCL9, CXCL10 and CXCL11 mRNA levels in FLI1-KO and WT groups of humanized NSG mice. **(3D)** HLA-A, HLA-B and HLA-C mRNA levels in FLI1-KO and WT groups of humanized NSG mice.

13) MC38 tumors respond very well to immunotherapy and even though interventions have been successful here they didn't always lead to successful therapies. Can these exciting results be confirmed in an additional model?

Response

We appreciate the reviewer's suggestion for reconfirming the role of FLI1 with an additional model. As mentioned above, we observe similar results (Figure R3.1A-1H) in an in-vivo model of NPC, which indicates intervention of FLI1 expression as a potential adjuvant to anti-PD-1 therapy.

14) The font of the words in the graphical abstract is too small.

Response:

According to the reviewer's suggestion, we have revised the font size of words.

15) If one looks at these data, it is surprising that IDO1 inhibitors failed in clinical trials, this should be discussed including possible compensatory mechanisms such as e.g. IL4I1, which also activates the AHR and is regulated in a similar manner

to IDO1.

Response:

We thank the reviewer for directing our attention to recent studies revealing IL411 as a compensatory mechanism of AHR activation that may explain failure of IDO1 inhibitor in clinical trials. We have thoroughly examined this publication and included a discussion of its findings in our revised manuscript (**see page 12, line 345 to line 348**).

Our point-by-point responses to the reviewer' comments are below.

Reviewer #1 (Remarks to the Author):

The authors have addressed all of my comments and I recommend to accept the paper.

Response to R1:

Thank you again for the positive comments and support for our work.

Reviewer #2 (Remarks to the Author):

All my concerns have been adequately addressed, and in my opinion this study is now suitable for publication in Nature Communications.

Response to R2:

Thank you again for the positive comments and support for our work.

Reviewer #3 (Remarks to the Author):

The authors have addressed my concerns.

Response to R3:

Thank you again for the positive comments and support for our work.